# Comparative Study on Production Performance of Different Oat (*Avena sativa*) Varieties and Soil Physicochemical Properties in Qaidam Basin

**DOI:** 10.3390/plants14131978

**Published:** 2025-06-28

**Authors:** Wenqi Wu, Ronglin Ge, Jie Wang, Xiaoli Wei, Yuanyuan Zhao, Xiaojian Pu, Chengti Xu

**Affiliations:** 1Academy of Animal Science and Veterinary Medicine, Qinghai University, Xining 810016, China; 18756758679@163.com (W.W.); 13594876904@163.com (R.G.); wangjie08142023@163.com (J.W.); 18894499178@163.com (X.W.); 18893147262@163.com (Y.Z.); puxj@qhu.edu.cn (X.P.); 2Key Laboratory of Northwest Cultivated Land Conservation and Marginal Land Improvement Enterprises, Ministry of Agriculture and Rural Affairs, Delingha 817000, China

**Keywords:** Qaidam Basin, saline-alkali land, oats, production performance, soil physicochemical properties

## Abstract

Oats (*Avena sativa* L.) are forage grasses moderately tolerant to saline-alkali soil and are widely used for the improvement and utilization of saline-alkali land. Using the oat varieties collected from the Qaidam Basin as experimental materials, based on the analysis data of the main agronomic traits, quality, and soil physical and chemical properties of different oat varieties at the harvest stage. The hay yield of Molasses (17,933.33 kg·hm^−2^) was the highest (*p* < 0.05), the plant height (113.59 cm) and crude fat (3.02%) of Qinghai 444 were the highest (*p* < 0.05), the fresh-dry ratio (2.62), crude protein (7.43%), and total salt content in plants (68.33 g·kg^−1^) of Qingtian No. 1 were the highest (*p* < 0.05), and the Relative forage value (RFV) of Baler (122.96) was the highest (*p* < 0.05). In the 0–15 cm and 15–30 cm soil layers of different oat varieties, the contents of pH, EC, total salt, Ca^2+^, Mg^2+^, and HCO_3_^−^ showed a decreasing trend at the harvest stage compared to the seedling stage, while the contents of organic matter, total nitrogen, Cl^−^, and SO4^2−^ showed an increasing trend. The contents of K^+^ and Na^+^ maintained a relatively balanced relationship between the seedling stage and the harvest stage in the two soil layers. Qingtian No. 1, Qingyin No. 1, and Molasses all rank among the top three in terms of production performance and soil physical and chemical properties, and they are the oat varieties suitable for cultivation in the research area.

## 1. Introduction

Soil salinization is one of the major abiotic stresses affecting crop production in the world, especially in arid and semi-arid regions [1,2,3]. Soil salinization has become a critical challenge for agriculture, significantly hindering agricultural development. The global extent of saline-alkali land is estimated to be approximately 800 million hectares [4]. It is estimated that the global annual crop yield loss caused by saline-alkali soil amounts to approximately USD 27.3 billion [5]. In recent years, the area of saline-alkali land in China has been gradually increasing, highlighting the urgent need for its development, utilization, and improvement. Soil salinity poses a significant threat to crop growth and yield, thereby hindering the sustainable development of modern agriculture [6]. Qinghai Province encompasses extensive areas of saline-alkali land, totaling approximately 1.13 million hectares. Notably, the Qaidam Basin in western Qinghai accounts for 1.02 million hectares, representing approximately 90% of the total area [7,8]. The average elevation of the Qaidam Basin is about 2600–3000 m, which is the highest inland basin in the world [9]. Soil salinization disrupts the granular structure of the soil, accelerates compaction, reduces the area of arable land, and adversely impacts crop growth, yield, and quality [10,11]. Under saline conditions, physiological and metabolic activities of plants are compromised due to osmotic stress, ionic stress, nutritional imbalances, or a combination of these factors [12,13,14]. Osmotic imbalance causes water deficit, reduced leaf area expansion, and stomatal closure which ultimately lessen photosynthesis and growth [15]. The ionic stress causes the excess accumulation of Na^+^ in the older leaves which leads to premature senescence of salt-accumulated older leaves [16]. Salinity significantly impacts key physiological processes in plants, including protein synthesis, energy metabolism, lipid metabolism during developmental stages, as well as cell wall biosynthesis and composition [17,18]. Photosynthesis, widely recognized as a vital biological process for the survival of plants, is significantly influenced by drought stress. Under drought conditions, there is a gradual decline in CO_2_ assimilation rates, accompanied by reductions in leaf size, stem elongation, and root proliferation. These changes disrupt plant water relations, diminish water-use efficiency, impair photosynthetic pigments, and adversely affect gas exchange, ultimately harming the plants [19]. In addition to drought stress, low temperatures are one of the most detrimental environmental stresses encountered by higher plants. Low temperature stress not only affects the growth and development of plants, but also significantly restricts their geographical distribution [20]. Oats (*Avena sativa* L.) enhance cold resistance through genetic regulation and metabolic adjustment. Field screening revealed that Genotypes CIav 4390 showed a 20% and 35% improvement over the standard checks, Okay and Dallas, respectively. Their cold resistance is closely related to the genotype-environment interaction [21]. Studies have shown that soil salinization is accelerating in arid and semi-arid areas. The Qaidam area is an extremely arid area with low annual precipitation and high salinity. The problem of soil salinization is becoming more and more serious, which has affected about 42.5% of the agricultural land in the basin [22]. The Qaidam region is characterized by a limited variety of forage species suitable for cultivation, as well as challenges related to seedling emergence and preservation. Additionally, the region faces slow growth rates, low yields, and low nutritional value of forage species, compounded by significant issues of soil salinity and alkalinity. [23]. Consequently, there is an urgent need to evaluate and identify salt-tolerant forage germplasm resources for saline-alkali land. Salt-tolerant plant resources hold great potential for expanding the utilization of saline-alkali land [24], facilitating a virtuous ecological cycle, and promoting sustainable agricultural production. Utilizing saline-alkali land for forage production represents the most ecologically and economically advantageous strategy, fostering the development of these areas as vital arable land resources and enhancing animal husbandry [25,26].

Oats are an annual dual-purpose crop belonging to the Gramineae family, utilized for both grain and forage [27]. They are widely cultivated in areas characterized by low fertility or saline-alkali conditions. Significant producers of oats include Northern Europe, North America, Australia, Canada, and China [28,29]. China is one of the world’s largest consumers of feed crops. The imports of oat hay from the United States, Australia, and other countries have steadily increased from 150,000 tons in 2008 to 390,000 tons in 2017 [27]. They are rich in essential nutrients, including fats, proteins, and vitamins [30,31], and are noted for their cold resistance, drought tolerance, and salt-alkali tolerance [32,33]. Harvested oats are primarily processed into forage products, such as silage, for livestock consumption, which enhances milk production, meat quality, and animal immunity [34]. As one of the more salt-tolerant crops within the grass family, oats are widely recognized as a viable option for the amelioration of saline soils [32,35]. The growth of oat plants is also affected by saline-alkali stress like other gramineous crops. The oat yield decreased by about 70% when the soil salt content was 0.6% [36]. There exists a significant variation in salt tolerance among different oat cultivar varieties [37], and the degree of their saline-alkali tolerance is closely associated with the inherent adaptability of the varieties themselves. Severe saline-alkali stress can inhibit the growth and development of oats, resulting in decreased yield and quality [38]. Relevant studies have also confirmed that the yield of crops planted under non-saline-alkali soil is much higher than that of crops planted under saline-alkali soil [39]. Therefore, saline-alkali stress is still an important limiting factor for oat growth, development, yield, and quality. In recent years, systematic progress has been made in the study of the molecular and biochemical mechanisms of salt tolerance in oats. The biochemical mechanisms mainly involve the synergy of antioxidant defense systems, the pH regulatory function of organic acid metabolism, and the dynamic balance of osmotic adjustment substances [40]. Research has shown that the *PeNAC1* gene significantly enhances the salt tolerance of oats by activating β-glucan synthesis genes (such as *AsCslF3* and *AsCslF6*). This gene regulates the synthesis of the cell wall component β-glucan, enhancing the stability of cell structure while promoting selective ion absorption by the roots [41]. In terms of adaptive regulation of energy metabolism pathways, the salt-tolerant variety Vao-9 significantly activates the pentose phosphate pathway and glycolysis under salt stress, while reducing energy consumption by inhibiting ribosome biogenesis, thereby enhancing the expression of heat shock proteins (*HSP70*, *HSP90*) and calreticulin, stabilizing protein folding, and alleviating oxidative damage [32]. The above studies indicate that oats adapt to salt stress through complex molecular and physiological-biochemical processes, thereby maintaining a dynamic balance in various aspects of the plant. The potential cultivation areas for oats are widely distributed across the Tibetan Plateau, and cultivated oats exhibit regional adaptability [42]. Therefore, it is crucial to conduct specific regional evaluation trials for different cultivated oat varieties. Studies have reported that experiments on the introduction and evaluation of oats have been conducted in various regions of Qinghai, China. These studies primarily focus on screening oat varieties suitable for cultivation in different ecological zones, emphasizing hay yield and nutritional quality. However, the majority of this research is concentrated in the eastern agricultural area, the Qinghai lake area, and the Three Rivers source area, and there are few reports on the introduction and evaluation of oats in the saline-alkali land of the Qaidam Basin [43]. Few studies have focused on selecting oat varieties suitable for cultivation in the study area by combining the production performance of cultivated oats in the growth period with the physical and chemical properties of soil, both domestically and internationally.

Saline alkaline land severely restricts the development of animal husbandry in the study area. Establishing artificial grasslands can alleviate ecological pressure, curb grassland degradation, and yield a significant amount of high-quality forage, thereby improving the issue of competition between livestock and humans for grain. The purpose of this experiment was to evaluate the growth performance of 15 oat varieties commonly cultivated in Qinghai Province, specifically in the saline-alkali land of the Qaidam Basin. Additionally, the study aimed to assess the changes in soil physical and chemical properties. Key indicators, including agronomic traits, nutritional quality, and soil physical and chemical properties, were analyzed. The test objectives were integrated with regional characteristics, focusing on two dimensions: variety adaptability and the physical and chemical properties of saline-alkali soil. This approach considers both scientific verification and production guidance. An evaluation system that integrates the production performance of cultivated oat varieties with the soil’s physical and chemical properties has been established in the saline-alkali soil of the Qaidam Basin. This approach overcomes the limitations of traditional single-yield screening methods. This study aims to identify oat varieties suitable for cultivation in saline-alkali land within the Qaidam Basin. Additionally, it seeks to determine the impact of soil physical and chemical properties on oat plants, thereby providing a scientific basis and theoretical reference for agricultural production.

## 2. Results

### 2.1. The Phenological Stages of Different Oat Varieties

All 15 varieties exhibited normal germination in the saline-alkali soil of the Qaidam Basin, with seedlings emerging 15 days after sowing. Notably, Mengshi No. 1 demonstrated the earliest germination, emerging just 8 days post-sowing. Furthermore, all tested oat varieties were suitable for forage harvesting during the designated period (Appendix A).

### 2.2. Comparison of Grass Yield Among Different Oat Varieties

Significant differences (*p* < 0.01) were observed in both fresh and hay yields among the various oat varieties. The average fresh yield for the 15 oat varieties at harvest was 28,508.89 kg·hm^−2^. The highest fresh yield was recorded in the Molasses variety, which reached 46,466.67 kg·hm^−2^, significantly surpassing the yields of other varieties during the same period (*p* < 0.05). In contrast, the lowest fresh yield was noted in Qinghai 444 at 16,300.00 kg·hm^−2^, which was significantly lower than those of other varieties (*p* < 0.05), with the exception of Gaoyan No. 1 and Haymaker (Figure 1a).

The average hay yield for the 15 tested oat varieties at harvest was 11,742.22 kg·hm^−2^, with values ranging from 7066.67 to 17,933.33 kg·hm^−2^ (Figure 1b). Among these varieties, Molasses exhibited the highest hay yield at 17,933.33 kg·hm^−2^, which was significantly greater than that of the other 14 oat varieties (*p* < 0.05).

### 2.3. Comparison of Agronomic Traits Among Different Oat Varieties

There were highly significant differences in plant height among various oat varieties at the harvest stage (*p* < 0.05) (Figure 2a). The average plant height across the 15 oat varieties was 105.59 cm. The tallest varieties included Qinghai 444 (113.59 cm), followed by Qingyin No. 1 (112.98 cm), Qinghai Sweet Oat (110.26 cm), and Baiyan No. 7 (110.21 cm). In contrast, Linna (99.67 cm), Mengshi No. 1 (99.10 cm), and Haymaker (89.74 cm) exhibited plant heights below 100 cm, indicating that these three varieties had relatively lower heights.

The average length, width, and leaf area of the flag leaves of the tested varieties were 26.57 cm, 1.77 cm, and 40.01 cm^2^, respectively. Among the varieties, the flag leaf of Da Fuweng was the longest at 31.82 cm, which was significantly greater than those of the other varieties (*p* < 0.05), except for Gao Yan No. 1 (28.76 cm) (Figure 2b). The flag leaf of Qinghai Sweet Oat was the widest at 2.23 cm, significantly exceeding those of the other varieties (*p* < 0.05), except for Gao Yan No. 1 (2.18 cm) (Figure 2c). The flag leaf area of Gao Yan No. 1 was the largest at 53.05 cm^2^, significantly greater than that of other varieties (*p* < 0.05), with the exception of Da Fuweng (52.74 cm^2^) and Qinghai sweet oat (52.39 cm^2^) (Figure 2d). These findings indicated that longer leaf length and wider leaf width significantly contribute to the formation of leaf area.

The number of leaves among the fifteen tested oat varieties ranged from 6.50 to 18.90. Qinghai sweet oat exhibited the highest leaf count at 18.90, which was significantly greater than that of the other varieties (*p* < 0.05), except for Gaoyan No. 1 (17.80) and Baler (16.30) (Figure 2e). The stem diameter across the fifteen oat varieties varied from 0.58 to 0.82 cm, with Qinghai sweet oat possessing the thickest stem at 0.82 cm, significantly surpassing the other tested varieties (*p* < 0.05) (Figure 2f). The number of tillers per plant among the tested oats ranged from 0.30 to 3.10, with Gaoyan No. 1 demonstrating the highest number of tillers at 3.10, significantly exceeding that of other varieties (*p* < 0.05), with the exception of Qinghai sweet oat (2.80) and Beile (2.30) (Figure 2g). The average fresh-to-dry ratio was 2.41, with Qingtian No. 1 exhibiting the highest ratio at 2.62, significantly higher than that of other oat varieties (*p* < 0.05), except for Molasses (2.59), Qingyin No. 2 (2.59), Qingyin No. 1 (2.58), Northwest No. 1 (2.52), Mengshi No. 1 (2.52), and Baler (2.42) (Figure 2h).

### 2.4. Comparison of Total Salt Content and Salt Accumulation

The data regarding the total salt content in the plants of the 15 tested oat varieties indicate that there is no significant difference in total salt content among the various oat varieties at the harvest period (*p* > 0.05). At this period, Qingtian No. 1 exhibited the highest total salt content in its plants, measuring 68.33 g·kg^−1^, followed closely by Qinghai sweet oat at 67.00 g·kg^−1^. The total salt content in these two varieties was significantly higher than that of Linna (46.00 g·kg^−1^) and Mengshi No. 1 (48.33 g·kg^−1^) (Figure 3a) (followed by Beile at 901.93 kg·hm^−2^). In contrast, Qinghai 444 exhibited the lowest accumulation at 412.03 kg·hm^−2^ (Figure 3b).

### 2.5. Comparison of Nutritional Quality and Relative Feed Value

The average crude protein (CP) content of the 15 tested oat varieties at harvest was 5.74%. Among these varieties, Qingtian No. 1 exhibited the highest CP content at 7.43%, significantly surpassing Dafuweng (4.55%), Mengshi No. 1 (3.80%), and Gaoyan No. 1 (3.80%) (*p* < 0.05). Mengshi No. 1 and Gaoyan No. 1 both displayed the lowest CP content at 3.80% (Figure 4a). The ether extract (EE) content at harvest revealed highly significant differences among the varieties (*p* < 0.01), with values ranging from 1.36% to 3.02%. Qinghai 444 recorded the highest EE content at 3.02%, significantly higher than that of other oat varieties (*p* < 0.05), while Gaoyan 1 had the lowest EE content at 1.36%, significantly lower than that of the other varieties (*p* < 0.05) (Figure 4b). The acid detergent fiber (ADF) and neutral detergent fiber (NDF) contents among the 15 tested oat varieties also exhibited extremely significant differences (*p* < 0.01). The ADF content ranged from 27.23% to 37.00% (Figure 4c), with Qinghai 444 having the highest ADF content at 37.00%, while Beile recorded the lowest ADF content at 27.23%, which was significantly lower than that of all other varieties except for Molasses (30.17%) and Da Fuweng (29.77%) (*p* < 0.05). The low ADF content suggests that these oat varieties possess high feeding quality. The NDF content varied from 51.23% to 66.87% (Figure 4d), with Qingyin No. 2 having the highest NDF content at 66.87%, significantly higher than that of all other varieties except for Haymaker (65.50%), Northwest No. 1 (63.73%), and Mengshi No. 1 (63.73%) (*p* < 0.05), indicating that these varieties have lower feeding value compared to other tested oat varieties during the same period. Beile, which has an NDF content of 51.23%, exhibited the lowest NDF content among the tested oat varieties. This finding suggests that this oat variety is easily digestible and readily absorbed by livestock, thereby providing a higher feeding value compared to the other oat varieties evaluated.

The analysis revealed highly significant differences in Relative forage value (RFV) among the 15 tested oat varieties (*p* < 0.01). The RFV values ranged from 87.31 to 122.96, with an average RFV of 99.48. Notably, the variety Beile exhibited the highest RFV at 122.96, which was significantly greater than all other oat varieties, with the exception of Gaoyan No. 1, which had an RFV of 114.66 (*p* < 0.05). Conversely, Haymaker recorded the lowest RFV at 87.31 (Figure 5).

### 2.6. Changes in Soil Physicochemical Properties of Different Oat Varieties

#### 2.6.1. The Impact of Different Oat Varieties on Soil pH

The effects of 15 oat varieties on soil pH reduction varied across different periods. Highly significant differences (*p* < 0.01) and significant differences (*p* < 0.05) in soil pH were observed between the control (CK) and the various oat varieties within the same soil layer (Table 1). During the seedling emergence and harvest periods, the soil pH values at a depth of 0–15 cm for the different oat varieties ranged from 8.30 to 8.81 and 7.88 to 8.60, respectively. The average soil pH at harvest decreased by 3.48% compared to the seedling stage. Notably, the soil pH of Northwest No. 1 decreased from 8.68 to 7.88, representing the largest reduction from seedling emergence to harvest at 9.22%. At the 15–30 cm depth, the soil pH values ranged from 8.44 to 8.85 and 7.83 to 8.80, respectively. The average soil pH at harvest decreased by 4.69% compared to the seedling stage. The soil pH of Linna decreased from 8.83 to 7.90, indicating a significant reduction of 10.53% from seedling emergence to harvest.

Throughout the growth period, from the seedling stage to the harvest stage, the soil pH in both the 0–15 cm and 15–30 cm layers of various oat varieties demonstrated a decreasing trend, whereas the control (CK) exhibited an increasing trend. Compared to CK during the same period, the average pH of the 0–15 cm soil layer at the seedling stage decreased by 2.44% across different oat varieties and by 7.01% at the harvest stage.

The average pH of the 15–30 cm soil layer decreased by 2.35% during the seedling stage and by 9.48% at the harvest stage. The lowest pH values in the 0–15 cm soil layer at both the seedling and harvest stages were recorded for Qinghai Sweet Oat and Northwest No. 1, with values of 8.30 and 7.88, respectively, indicating reductions of 5.79% and 11.66% compared to the CK. These two varieties exhibited the most significant pH reductions in the 0–15 cm soil layer relative to CK. In the 15–30 cm soil layer, the lowest pH values at both stages were observed for Northwest No. 1, measuring 8.44 and 7.83, respectively, which correspond to reductions of 5.27% and 14.52% compared to CK. This variety demonstrated the greatest pH reduction in the 15–30 cm soil layer compared to CK.

#### 2.6.2. The Impact of Different Oat Varieties on Soil EC

There were highly significant differences in soil electrical conductivity between the CK and various oat varieties within the same soil layer (*p* < 0.01) (Table 2). The soil electrical conductivity (EC) of different oat varieties during the emergence and harvest periods at a depth of 0–15 cm ranged from 0.23 to 0.63 ms/m and from 0.17 to 0.47 ms/m, respectively. The average soil EC of different oat varieties at harvest decreased by 32.87% compared to the seedling stage, with Jiayan No. 2 exhibiting the most substantial reduction, from 0.53 ms/m to 0.17 ms/m, a decrease of 67.92%. At a depth of 15–30 cm, the soil EC ranged from 0.27 to 0.77 ms/m and from 0.13 to 0.47 ms/m during the emergence and harvest periods, respectively. The average soil EC at harvest also decreased by 33.51% compared to the seedling stage, with Molasses showing the largest decline, from 0.70 ms/m to 0.13 ms/m, indicating a reduction of 81.43%.

Throughout the growth period, from the seedling stage to the harvest stage, the soil EC at depths of 0–15 cm and 15–30 cm for various oat varieties demonstrated a decreasing trend, whereas the CK exhibited an increasing trend. Specifically, during the emergence stage, the average soil EC at 0–15 cm decreased by 37.71% compared to CK, while during the harvest stage, this decrease was 63.81%. For each oat variety, the average soil EC at 15–30 cm during the emergence stage decreased by 41.42% relative to CK, and during the harvest stage, it decreased by 65.38%. Notably, the lowest soil EC at 0–15 cm was recorded in Molasses and Jiayan No. 2, with values of 0.23 ms/m and 0.17 ms/m, respectively, indicating reductions of 67.14% and 77.92% compared to CK. These two varieties exhibited the most significant reductions in soil EC relative to CK. Similarly, the lowest soil EC at 15–30 cm was found in Qingyin No. 1 and Molasses, with values of 0.27 ms/m and 0.13 ms/m, respectively, reflecting reductions of 66.25% and 84.34% compared to CK. These two varieties also showed the most pronounced reductions in soil EC compared to CK.

#### 2.6.3. The Impact of Different Oat Varieties on Total Soil Salinity

A highly significant difference (*p* < 0.01) was observed in the total soil salt content between the CK and various oat varieties within the same soil layer (Table 3). The total soil salt content at a depth of 0–15 cm during the seedling emergence and harvest periods for the oat varieties ranged from 0.61 to 2.27 g·kg^−1^ and 0.36 to 1.36 g·kg^−1^, respectively. Across all oat varieties, the total soil salinity in the 0–15 cm layer decreased by an average of 40.14% from the seedling stage to the harvest stage. Notably, the total soil salinity of Haymaker decreased from 2.27 g·kg^−1^ to 0.36 g·kg^−1^, representing the largest reduction from the seedling stage to the harvest stage at 84.14%. In the 15–30 cm layer, the total soil salinity for all oat varieties at the seedling and harvest stages ranged from 0.75 to 2.18 g·kg^−1^ and 0.31 to 1.73 g·kg^−1^, respectively. The average decrease in total soil salinity in this layer across all oat varieties was 44.58% from the seedling stage to the harvest stage. Among these, the total soil salinity of Qinghai sweet oat decreased from 1.49 g·kg^−1^ to 0.31 g·kg^−1^, indicating a significant reduction of 79.19% from the seedling stage to the harvest stage.

From the seedling stage to the harvest stage, as the growth period progresses, the total soil salt content in the 0–15 cm and 15–30 cm layers of different oat varieties shows a decreasing trend compared to the harvest stage, while the CK exhibits an increasing trend. Compared to the total salt content in CK soil during the same period, the average total salt content in the 0–15 cm soil layer during the seedling emergence stage decreased by 37.12%, and during the harvest stage, it decreased by 66.77%. In the 15–30 cm soil layer, the average total salt content during the seedling emergence stage decreased by 37.84%, and during the harvest stage, it decreased by 67.90%. During both the seedling emergence and harvest stages, the lowest total salt content in the 0–15 cm soil layer was observed in the Molasses and Haymaker varieties, with values of 0.61 g·kg^−1^ and 0.36 g·kg^−1^, respectively, representing reductions of 75.10% and 85.77% compared to CK. These two varieties exhibited the most significant reduction in total soil salt content compared to CK. During the seedling emergence stage, the lowest total salt content in the 15–30 cm soil layer was recorded in Qingtian No. 1, at 0.75 g·kg^−1^, reflecting a decrease of 68.49% compared to CK, which was the largest reduction among all oat varieties. During the harvest stage, the lowest total salt content in the 15–30 cm soil layer was observed in Qinghai sweet oat and Qingyin No. 1, both at 0.31 g·kg^−1^, representing an 87.30% reduction compared to CK. These two varieties also exhibited the most significant reduction in total soil salt content compared to CK.

#### 2.6.4. The Impact of Different Oat Varieties on Soil Nutrient Characteristics

The effects of various oat varieties on soil nutrients are presented in Appendix A. It is evident that the nutrient content in the 0–15 cm soil layer surpasses that in the 15–30 cm layer across different growth stages, with the harvest period demonstrating superior values compared to the seedling stage. As planting time progresses, all nutrient indicators show an upward trend. There are highly significant differences (*p* < 0.01) in soil organic matter and total nitrogen between the CK and the different oat varieties within the same soil layer. At harvest, the soil organic matter and total nitrogen content in the 0–15 cm layer of the 15 tested oat varieties increased by an average of 34.20% and 28.51%, respectively, compared to the seedling stage. Notably, Northwest No. 1 exhibited the most substantial increase in soil organic matter content, rising from 8.26 g·kg^−1^ to 17.23 g·kg^−1^, which corresponds to an increase of 108.60%. Mulesi demonstrated the highest increase in total nitrogen content, climbing from 0.41 g·kg^−1^ to 0.67 g·kg^−1^, a rise of 63.41%. In the 15–30 cm soil layer during the harvest period, the organic matter and total nitrogen content increased by an average of 23.91% and 25.38%, respectively, compared to the seedling stage. Among these, Qingtian No. 1 showed the most significant increase in soil organic matter content, from 6.32 g·kg^−1^ to 9.88 g·kg^−1^, reflecting a 56.33% increase. Qingyin No. 1 exhibited the highest increase in total nitrogen content, rising from 0.34 g·kg^−1^ to 0.62 g·kg^−1^, an increase of 82.35%.

During a single growth cycle, the soil organic matter content of various oat varieties exhibited an upward trend within the same soil layer as the growth period progressed (Appendix A). At the seedling stage, the soil organic matter contents of Jiayan No. 2 (7.48 g·kg^−1^) and Muwang (6.72 g·kg^−1^) in the 0–15 cm soil layer were lower than that of the CK, (7.61 g·kg^−1^), although these differences were not statistically significant (*p* > 0.05). In contrast, the soil organic matter contents of other tested oat varieties were all greater than that of CK (7.61 g·kg^−1^), resulting in an overall average soil organic matter content that was 34.88% higher than CK. At the harvest stage, the soil organic matter contents of all 15 tested oat varieties in the 0–15 cm soil layer exceeded that of CK (8.03 g·kg^−1^), with an average increase of 67.75% over CK. During the seedling stage, the soil organic matter contents of the 15 tested oat varieties in the 15–30 cm soil layer were also higher than those of CK (4.08 g·kg^−1^); however, no significant differences were observed between Linna (4.45 g·kg^−1^), Jiayan No. 2 (4.51 g·kg^−1^), and CK (4.08 g·kg^−1^) (*p* > 0.05). The overall average soil organic matter of the different oat varieties was 48.79% higher than that of CK. At the harvest stage, the soil organic matter contents of the 15 tested oat varieties in the 15–30 cm soil layer were again higher than those of CK (3.77 g·kg^−1^), with an overall average increase of 99.38% compared to CK.

The trend of total nitrogen content in soil across different oat varieties throughout the growth cycle was consistent. Among the 15 tested oat varieties, the total nitrogen content in the 0–15 cm and 15–30 cm soil layers exhibited an increasing trend as the growth period progressed. Notably, the total nitrogen content in the 15–30 cm soil layer was lower than that in the 0–15 cm soil layer (Appendix A). During the seedling stage, the total nitrogen content in the 0–15 cm soil layer of the Mulesi variety (0.41 g·kg^−1^) was the lowest, being 16.33% lower than that of the CK variety (0.49 g·kg^−1^). However, the overall average total nitrogen content in the soil was 11.02% higher than that of CK. Among the tested varieties, Qinghai sweet oats exhibited the highest total nitrogen content in the soil at 0.69 g·kg^−1^, which was 40.82% higher than that of CK. At the harvest stage, the total nitrogen content in the 0–15 cm soil layer of all tested oat varieties surpassed that of CK (0.53 g·kg^−1^). Except for Linna (0.58 g·kg^−1^), where the difference in total nitrogen content compared to CK (0.53 g·kg^−1^) was not significant (*p* > 0.05), the total nitrogen content in the soil of the other oat varieties was significantly higher than that of CK (*p* < 0.01). The average total nitrogen content in the soil across different oat varieties was 29.69% higher than that of CK. During the seedling stage, the total nitrogen content in the 15–30 cm soil layer of CK was the lowest at 0.33 g·kg^−1^, while the average total nitrogen content in the soil of the 15 tested oat varieties was 31.92% higher than that of CK. Similarly, during the harvest period, the total nitrogen content in the 15–30 cm soil layer of CK remained the lowest at 0.37 g·kg^−1^, and the average total nitrogen content in the soil of the 15 tested oat varieties was 46.13% higher than that of CK.

#### 2.6.5. The Effects of Different Oat Varieties on Soil Base Ions

The cultivation of different oat varieties has generally resulted in a reduction of soil Ca^2+^, Mg^2+^, and HCO_3_^−^ content (Appendix A). In contrast, the levels of K^+^ and Na^+^ remained relatively balanced during the two growth periods (Appendix A). Additionally, there was an overall increase in soil Cl^−^ and SO_4_^2−^ content (Appendix A). At the harvest stage, the average reductions in Ca^2+^, Mg^2+^, and HCO_3_^−^ content in the 0–15 cm soil layer of various oat varieties were 41.30%, 34.73%, and 19.34%, respectively, when compared to the content at the emergence stage. Similarly, in the 15–30 cm soil layer, the average decreases in Ca^2+^, Mg^2+^, and HCO_3_^−^ content at the harvest stage were 27.33%, 34.20%, and 15.85%, respectively, when compared to the seedling stage. Furthermore, at the seedling stage, the average contents of Ca^2+^, Mg^2+^, and HCO_3_^−^ in the 0–15 cm soil layer of different oat varieties were 58.06%, 48.15%, and 23.01% lower than those in the CK soil.

At the harvest stage, the concentrations of Ca^2+^, Mg^2+^, and HCO_3_^−^ in the 0–15 cm soil layer of the 15 tested oat varieties decreased by an average of 71.79%, 74.67%, and 13.33%, respectively, compared to the CK soil. During the seedling stage, the concentrations of Ca^2+^, Mg^2+^, and HCO_3_^−^ in the 15–30 cm soil layer decreased by an average of 55.95%, 56.00%, and 15.90% compared to CK. At the harvest stage, the concentrations of Ca^2+^ and Mg^2+^ in the 15–30 cm soil layer decreased by an average of 63.33% and 65.00%, respectively, compared to the CK soil, while HCO_3_^−^ increased by 2.22%. Additionally, at the harvest stage, the concentrations of K^+^, Na^+^, Cl^−^, and SO_4_^2−^ in the 0–15 cm soil layer for different oat varieties increased by an average of 21.49%, 116.62%, 199.67%, and 161.21%, respectively, compared to the seedling stage.

The concentrations of K^+^, Na^+^, Cl^−^, and SO_4_^2−^ in the 15–30 cm soil layer of different oat varieties at the harvest stage increased by an average of 13.46%, 32.74%, 99.40%, and 23.38%, respectively, compared to the seedling stage. The 15 tested oat varieties exhibited an average reduction of 51.93%, 75.68%, 61.52%, and 61.73% in soil K^+^, Na^+^, Cl^−^, and SO_4_^2−^ concentrations at a depth of 0–15 cm during the emergence stage when compared to the CK soil. During the harvest stage, the average reductions in soil K^+^, Na^+^, Cl^−^, and SO_4_^2−^ concentrations at the same depth were 49.02%, 61.95%, 37.73%, and 46.67%, respectively, compared to CK. At a depth of 15–30 cm during the emergence stage, the average reductions in soil K^+^, Na^+^, Cl^−^, and SO_4_^2−^ concentrations were 50.37%, 57.71%, 62.38%, and 54.62%, respectively, compared to the CK soil. During the harvest stage, the average reductions in soil K^+^, Na^+^, Cl^−^, and SO_4_^2−^ concentrations at 15–30 cm depth were 50.00%, 53.12%, 57.47%, and 53.92%, respectively, compared to the CK soil.

### 2.7. Correlation Analysis of Cultivated Oat Production Performance and Soil Physical and Chemical Properties

Pearson correlation analysis (PCA) of agronomic traits, nutritional quality, and soil physical and chemical properties of 15 oat varieties is shown in Figure 6. The figure illustrates a correlation between agronomic traits, nutritional quality, and soil physical and chemical properties. There was a significant positive correlation between the fresh grass yield and the hay yield of cultivated oats (*p* < 0.01). Fresh grass yield was significantly negatively correlated with soil Ca^2+^, Na^2+^, and Cl^−^ (*p* < 0.05), and was significantly negatively correlated with SO_4_^2−^ (*p* < 0.01). Hay yield was significantly negatively correlated with soil Na^+^, Cl^−^ and SO_4_^2−^ (*p* < 0.05). Soil pH was significantly negatively correlated with soil total salt and Mg^2+^ (*p* < 0.05), and significantly positively correlated with HCO_3_^−^ (*p* < 0.05). Soil electrical conductivity was significantly positively correlated with soil total salt (*p* < 0.05) and extremely significantly positively correlated with Ca^2+^ (*p*< 0.01). Soil organic matter was extremely significantly positively correlated with soil total nitrogen (*p* < 0.01).

### 2.8. Comprehensive Evaluation of Production Performance and Soil Physicochemical Properties of Different Oat Varieties

#### 2.8.1. PCA and Membership Function

A PCA was conducted on 13 indicators of 15 oat varieties during the harvest period (Table 4). The results showed that the contribution rates of the first four principal components were 36.496%, 21.094%, 15.664%, and 12.290%, respectively. The cumulative contribution rate of these four principal components reached 85.544%, which essentially represents 85.544% of the information of the tested oat varieties. Therefore, these four principal components can be used as the main factors for a comprehensive evaluation of the production performance of the 15 oat varieties. By analyzing the eigenvectors of different comprehensive indicators, it can be seen that in the information represented by principal component 1, the eigenvectors for tiller number, stem diameter, fresh-to-dry ratio, flag leaf area, RFV, and NDF are larger, indicating that these six indicators play a major role in principal component 1; in principal component 2, the eigenvectors for hay yield, fresh grass yield, and ADF are larger; in principal component 3, the eigenvector for total salt content in plants is larger; and in principal component 4, the eigenvectors for CP and EE are larger. In summary, through principal component analysis, 13 performance-related indicators of 15 oat varieties were converted into 4 independent comprehensive indicators for the next step of comprehensive performance evaluation analysis.

According to Formula (5), the membership function values of the four principal components for the 15 oat varieties were obtained. By combining the contribution rates of the four principal components, the weights of the four principal components were calculated as 0.4266, 0.2466, 0.1831, and 0.1437 according to Formula (6). Using Formula (7), the comprehensive evaluation value D for the production performance of 15 oat varieties was calculated. The larger the D value, the stronger the production performance. Based on the comprehensive evaluation D value, the comprehensive ranking of the production performance of the 15 oat varieties at the harvest period was obtained (Table 5). The results showed Qingtian No. 1 > Qingyin No. 1 > Molasses > Qingyin No. 2 > Northwest No. 1 > Jiayan No. 2 > Baler > Qinghai 444 > Qinghai sweet oat > Baiyan No. 7 > Haymaker > Mengshi No. 1 > Da Fuweng > Lena > Gaoyan No. 1. The results indicated that the three varieties, Qingtian No. 1, Qingyin No. 1, and Molasses, have demonstrated top-tier comprehensive performance in terms of production capabilities.

#### 2.8.2. Membership Function Values of Soil Physicochemical Properties

The results of this experiment indicate that different oat varieties have varying effects on the physicochemical properties of saline-alkali soil in the Qaidam Basin. Based on 12 indicators of soil physicochemical properties at the harvest stage, including pH, EC, organic matter, total nitrogen, total soil salt, Ca^2+^, Na^+^, K^+^, Mg^2+^, Cl^−^, HCO_3_^−^, and SO_4_^2−^ content, from 15 tested oat varieties, the membership function values of each indicator were calculated for a comprehensive evaluation (Appendix A). The results show: Qingtian No. 1 > Qingyin No. 1 > Molasses > Qinghai sweet oat > Da Fuweng > Qingyin No. 2 > Baler > Qinghai 444 > Baiyan No. 7 > Mengshi No. 1 > Jiayan No. 2 > Haymaker > Lena > Northwest No. 1 > Gaoyan No. 1. The results indicated that the three varieties, Qingtian No. 1, Qingyin No. 1, and Molasses, rank at the top in terms of comprehensive performance in soil physicochemical properties.

## 3. Discussion

The average annual precipitation in the Qaidam Basin is only 50–200 mm, making oats one of the few crops that can be cultivated on a large scale. Furthermore, as a high-quality forage, oats are relied upon by pastures surrounding the Qaidam Basin as winter feed [44]. The root system of oats can reduce the accumulation of salts in the topsoil by 30–40%, which is crucial for the Qaidam Basin [45]. The aforementioned studies indicate that cultivating oats in the Qaidam Basin holds significant importance and advantages. Zhou et al. proved that the salt-tolerant variety Qingyongjiu No. 195 possessed stronger salt tolerance than the sensitive variety 709, indicating that oats are salt-tolerant [46]. Fan et al. demonstrated that the growth of oats under low, medium, and high salt stress on saline-alkali soil can adapt to the saline-alkali conditions of soda-alkali soil to a certain extent; however, this adaptability decreases as the degree of stress increases [47]. Oats are recognized as a moderately salt-tolerant crop and can be utilized to enhance saline-alkali land [32]. Different oat varieties exhibit distinct genetic traits and varying degrees of environmental adaptability. The successful introduction of oats is primarily determined by the completeness of their growth and development. By observing the growth performance of a specific variety in a particular region, one can ascertain its suitability for the local planting environment. The introduction of oats to the saline-alkali land in the Qaidam Basin represents a novel endeavor. In this study, 15 oat varieties demonstrated normal growth and development in the saline-alkali land of the Qaidam Basin, indicating the feasibility of cultivating oats as an annual forage crop in this region. This paper assesses the application value of oats as forage and their resistance to salt stress by planting 15 different oat varieties in the saline-alkali land of the Qaidam Basin. Research indicated that under saline-alkali stress conditions, the yield of forage grass decreases compared to non-saline-alkali conditions. In this study, the yield of the Bale fresh grass variety was 34,800.00 kg·hm^−2^, with a hay yield of 14,433.33 kg·hm^−2^. Zhang et al. [48] reported that under non-saline-alkali conditions, the yield of the Bale Fresh Grass variety was 56,424.43 kg·hm^−2^, and the hay yield was 15,013.06 kg·hm^−2^. This comparison clearly indicated that saline alkali land results in a significant reduction in grass yield.

The Qingtian No. 1 plant exhibited the highest total salt content at 68.33 g·kg^−1^. Although the Mulesi plant had a lower total salt content of 51.67 g·kg^−1^, it achieved the highest hay yield at 17,933.33 kg·hm^−2^. Consequently, the salt accumulation in Mulesi reached 927.13 kg·hm^−2^ under identical conditions, surpassing that of Qinghai by 444 to 515.10 kg·hm^−2^. This finding suggests that a significant advantage in hay yield can facilitate substantial salt accumulation, highlighting the importance of hay yield in the context of salt accumulation.

Saline alkali land also affects the flag leaf length, width, area, and CP content of plants. For example, Li et al. [49] showed that the flag leaf lengths of the varieties Haymaker and Beile were 43.03 cm and 34.40 cm, respectively; the flag leaf widths were 2.50 cm and 2.00 cm, respectively; and the flag leaf areas were 89.54 cm^2^ and 57.25 cm^2^, respectively. This experimental study revealed that the flag leaf lengths of the varieties Haymaker and Beile were 26.65 cm and 25.41 cm, respectively; the flag leaf widths were 1.41 cm and 1.79 cm, respectively; and the flag leaf areas were 31.43 cm^2^ and 38.34 cm^2^, respectively. The flag leaf length, width, and area in the aforementioned study were superior to those of the varieties Haymaker and Beile in this experiment. Li et al. [49] indicated that the crude protein content of Haymaker and Beile was higher than 10%, whereas in this experiment, the crude protein contents of Haymaker and Beile were 5.58% and 6.13%, respectively, which were lower than those in the aforementioned study. The saline-alkali components present in saline-alkali soil exhibit physiological toxicity that adversely affects plant growth, impeding normal development. Furthermore, the production potential of saline-alkali soil is limited, failing to offer the nutrient conditions necessary for optimal plant growth and development, which in turn restricts crop growth and leads to reductions in flag leaf length, width, and area. Since the protein content in leaves is relatively high, the crude protein content of the varieties Haymaker and Beile in this experiment was also lower than that in the study by Li et al. [49]. RFV is a significant indicator for selecting high-quality forage by integrating the performance of ADF and NDF, widely used in the evaluation model of roughage feeding value. The higher the RFV, the greater the feeding value of the forage [50]. Li et al. [51] showed that the RFV of Qinghai sweet oats, Qingyin No. 2, and Baiyan No. 7 were 117.53, 124.37, and 125.51, respectively. In this experiment, the RFV of Qinghai sweet oats, Qingyin No. 2, and Baiyan No. 7 were 95.00, 87.55, and 98.28, respectively, which are lower than those in the study by Li et al. [51]. This may be due to the adverse effects of saline-alkali soil on crop quality.

The soil pH, EC, and total salt content of 15 oat varieties in the 0–15 cm and 15–30 cm soil layers showed a gradual decreasing trend over the growing period. From the seedling emergence stage to the harvest stage, different oat varieties demonstrated certain effectiveness in inhibiting soil salinization over time. The reduction in soil salinity was greater in the 15–30 cm soil layer compared to the 0–15 cm layer, specifically manifested in the greater decreases in pH, EC, and total salt content in the 15–30 cm layer than in the 0–15 cm layer. Related studies have shown that the root zone shows a decrease in total soil salt content [52]. On the one hand, roots actively promote the reduction of salt in the 15–30 cm soil layer through selective ion absorption and water extraction. In this depth range, plants preferentially absorb water and nutrients from soil patches with low salinization. On the other hand, the climate in the Qaidam area is dry, and the high temperature and little rain lead to evaporation and transpiration exceeding precipitation. Through capillary action, the salty water in the deep soil is absorbed into the dry surface soil. When water evaporates from the soil surface, the previously dissolved salts remain as undissolved solids. Due to insufficient water to dissolve or leach away the salts, accumulation of salt occurs in the surface soil [53,54,55]. When the soil EC reaches 4 dS/m or higher, the soil is classified as a saline soil. At such soil salinity levels, the growth and yield of most crops are significantly reduced [56]. Related studies have shown that with the increase of EC, all agronomic traits, including aboveground biomass, plant height, tiller number, and stem diameter, are negatively affected, and with the increase of EC, the impact is increasing [27]. Feeding oats grows vigorously when the conductivity is 2.0–4.5 dS/m. Above 6.0 dS/m, the relative aboveground biomass decreased, and photosynthetic efficiency declined due to chloroplast damage induced by sodium ions [27]. The soil electrical conductivity of the oat varieties cultivated in this study predominantly exceeded 4 dS/m, categorizing them as saline-alkali land. The study also observed a decrease in oat yield, which further confirmed that oats are moderately salt-tolerant crops. It was found that the total soil salt content of the cultivated oat varieties showed a decreasing trend and could be used to improve the saline-alkali land. Plant responses to salt stress are highly complex, involving interactions between physiological processes, metabolic pathways, and molecular and cellular regulation [14]. While oat salt tolerance has been studied at different levels, the underlying mechanisms remain poorly understood [46]. The mechanisms of salt tolerance are categorized into three types: primarily osmotic stress tolerance, ion exclusion, and tissue tolerance. In most conditions, plant salt stress tolerance relies on these three mechanisms collectively rather than on a single one [15,16,57]. In this study, the total salt content in the soil decreased compared to the CK, suggesting that the primary mechanism of salt tolerance might be osmotic stress tolerance, with the possible involvement of the other two mechanisms. Our hypothesis requires further experimental verification. The soil nutrient content (organic matter, total nitrogen, total phosphorus) in the 0–15 cm soil layer is superior to that in the 15–30 cm layer, and the harvest period outperforms the seedling stage. All nutrient indicators exhibited an upward trend as the planting period progressed. This phenomenon may be attributed to the ability of plant roots to recruit relevant microorganisms during growth, whereby the synergistic interaction between microorganisms and plants can effectively regulate the physicochemical properties of the soil. The soil nutrient content in the upper layer is higher than in the lower layer, possibly due to more intense microbial activity in the upper soil and relatively more plant humus, leading to an increase in soil nutrient content. The 15 oat varieties exhibited different trends in K^+^ and Na^+^ levels across the two soil layers, which may be attributed to competitive inhibition between Na^+^ and K^+^ uptake, consistent with the findings of Li et al. [58]. Related studies have also shown that there is a competitive inhibition between Na^+^ and K^+^ uptake under saline-alkali stress [58]. Salt stress triggers complex ion uptake and transport mechanisms in plants to maintain K^+^/Na^+^ homeostasis and mitigate Na^+^ toxicity. To survive salt stress, plants must maintain a balance between sodium (Na^+^) and potassium (K^+^) ions. High-affinity potassium transporters (HKTs) play a crucial role in mitigating Na^+^ toxicity by facilitating K^+^ uptake. The uptake of potassium is vital for plants in the context of salt stress [59,60]. High Na^+^ content in the cytosol leads to severe K^+^ deficiency. One strategy to counteract K^+^ deficiency is to activate high-affinity K^+^ transporters to take up K^+^ and thus maintain the ionic balance at the cell level [61,62]. This study found that under saline-alkali stress, the Ca^2+^ and Mg^2+^ contents of the 15 oat varieties also generally decreased, which is also in agreement with the results of Li et al. [58]. When plants require CO_2_ for photosynthesis, HCO_3_^−^ can be converted into CO_2_ for plant utilization, thereby participating in photosynthesis. As an essential component of the intracellular acid-base balance regulation system, HCO3^−^ can interact with H^+^ and others to maintain stable intracellular pH levels [63]. The high concentration of bicarbonate (HCO_3_^−^) in saline-alkali soil leads to an increase in soil pH value, which is mainly due to the hydrolysis of carbonate substances. The hydrolysis of bicarbonate ions will release hydroxide ions (OH^−^) into the soil solution. The generated hydroxide ions will increase the pH value of the soil and present a toxic alkaline environment for plants. These environments are caused by the presence of HCO_3_^−^. In alkaline soils with low solubility of carbon dioxide, this reaction is particularly significant, which will shift the equilibrium to the direction of generating hydroxide ions [64]. In this study, the content of HCO_3_^−^ ions in the soil decreased, and the absorption of HCO_3_^−^ by plants promoted plant growth, ultimately leading to a reduction in soil pH. A potential limitation of this study is the inability to determine the specific pathways through which oats resist saline-alkali stress. Therefore, we will next conduct further experimental verification of the specific adaptation mechanisms to elucidate the saline-alkali resistance mechanisms of oats.

## 4. Materials and Methods

### 4.1. Overview of the Experimental Site

The experimental site is located in Gahai Town, Delingha City, Qinghai Province, China (97°22′44′′ E, 37°15′13′′ N, altitude 2842.3 m), on the northeastern edge of the Qaidam Basin, characterized by a plateau mountain climate. The annual average temperature is 2.8 °C, with an annual accumulated temperature ≥ 0 °C of 2363.9 °C over an average of 216 d, and an annual accumulated temperature ≥ 10 °C of 660.0 °C over an average of 113 d. The frost-free period lasts between 90 and 110 d, with an annual total of 3182.8 h of sunshine. The annual solar radiation measures 693.33 kJ·cm^−2^, while the average annual precipitation is 181.8 mm, and the annual evaporation is 2370.0 mm. The experimental site consists of desert saline-alkali farmland, with soil classified as salinized cultivated and irrigated brown calcium soil. The basic nutrients of the soil in the test site are ammonium nitrogen 6.39 mg·kg^−1^, nitrate nitrogen 21.87 mg·kg^−1^, available phosphorus 28.08 mg·kg^−1^, and available potassium 129.61 mg·kg^−1^.

### 4.2. Test Materials

The 15 tested oat cultivar varieties are as follows: (A) Qingtian No. 1; (B) Molasses; (C) Qinghai sweet oat; (D) Qingyin No. 1; (E) Qingyin No. 2 (F) Dafuweng; (G) Baiyan No. 7; (H) Northwestern No. 1; (I) Lena; (J) Jiayan No. 2; (K) Mengshi No. 1; (L) Qinghai No. 444; (M) Haymaker (N) Gaoyan No. 1 (O) Baler. The above varieties were provided by the Academy of Animal Husbandry and Veterinary Sciences, Qinghai University.

### 4.3. Experimental Design and Methodology

The experiment was conducted using a randomized block design, with each plot measuring 30 m^2^ (5 m × 6 m) and comprising three replications. Prior to the trial, the land was plowed, leveled, and harrowed. Seeds were manually sown in furrows, maintaining a plot spacing of 0.5 m, a row spacing of 15 cm, and a sowing depth of 3 cm. The test oats were uniformly sown on 13 June 2023, at a seeding rate of 150 kg·hm^−2^, accompanied by conventional irrigation (250–300 m^3^ per mu). The base fertilizer application comprised organic fertilizer (with an N-P_2_O_5_-K_2_O ratio of 1.5–0.9–1.6 and a total nutrient content of ≥4.0%) at a rate of 15,000.00 kg·hm^−2^, along with diammonium phosphate at 300.00 kg·hm^−2^. The experimental area was safeguarded by fencing around its perimeter. Relevant indicators were measured from the seedling stage through to the harvest stage of the forage oats.

### 4.4. Measurement Indicators and Methods

#### 4.4.1. Phenological Stages and Agronomic Traits

The overall observation method was employed, with the criterion being that 50% of the plants in the experimental plots entered a specific phenological phase. This criterion was used to observe and record the emergence and harvest periods of each oat variety. During the harvest period, ten uniformly growing oat plants were randomly selected from each plot to measure the absolute plant height, tiller number, and leaf number of each plant. The diameter of the stem was measured at the thickest part of the second stem node using a vernier caliper [65]. The maximum length of the flag leaf on the main stem was measured with a ruler, referred to as the flag leaf length (L), while the maximum width of the flag leaf was measured as the flag leaf width (W). The area of the flag leaf (A) was calculated using the following formula:(1)A=R×L×W

In the formula, R is the correction coefficient for oat leaf area (0.8317) [66].

#### 4.4.2. Grass Yield and Total Salt Content in Plants

During the oat harvest period, the marginal effects were first eliminated within each plot. Subsequently, a representative sample area of 1 m^2^ was selected within the plot. After mowing, the fresh grass weight was measured to calculate the fresh grass yield. The oat samples were then placed in a drying room for natural air-drying to determine the hay weight, which was subsequently used to measure the biomass. The hay yield and the fresh-to-dry ratio were calculated accordingly. The air-dried samples were then crushed for further analysis [67]. The total salt content in the plant was represented by crude ash, and the crude ash content was determined using the muffle furnace ignition method (GB/T 6438-2007) [29].

#### 4.4.3. Nutritional Quality

During the harvest period, air-dried samples from each plot were ground, passed through a 40-mesh sieve, and subsequently stored for the determination of the nutritional quality of oats. The CP content was determined using the Kjeldahl method, while the ether extract (EE) content was measured using the Soxhlet fat extraction method. Additionally, the ADF and NDF contents were assessed using the filter bag method [68]. The formula for calculating the RFV is presented below [69].(2)DDM(%DM)=88.9−0.779×ADF(%DM)(3)DMI(%BW)=120/NDF(%DM)(4)RFV=DDM(%DM)×DMI(%BW)/1.29

In the formula, DDM represents digestible dry matter (%DM); DMI represents dry matter intake (%BW); DM represents dry matter; and BW represents body weight.

#### 4.4.4. Soil Measurement Indicators and Methods

Soil samples from the plow layer (0–15 cm and 15–30 cm) were collected at the seedling and harvest stages of each oat variety in each plot using a soil auger, following the five-point sampling method. The soil samples from the same layer were thoroughly mixed and transported to the laboratory, where root stubble and gravel were removed. After natural air-drying, the samples were ground and passed through an 80-mesh sieve. Relevant soil indicators were measured as follows: soil pH was determined using a pH meter (Mettler Toledo, Switzerland); soil electrical conductivity was measured using the electrode method [70]; soil organic matter content was assessed using the potassium dichromate oxidation external heating method [71]; total soil nitrogen content was determined via the Kjeldahl method [72]; the mass fraction of total soil salt content was determined using the drying method [67]; Ca^2+^ and Mg^2+^ were measured using the EDTA complexometric titration method [73]; HCO_3_^−^ was determined using the double indicator neutralization titration method [74]; Na^+^ and K^+^ were measured using the flame photometry method [75]; Cl^−^ was assessed using the standard AgNO_3_ titration method [74]; and SO_4_^2−^ was measured using the EDTA indirect complexometric titration method [76]. Salt accumulation in oats was calculated as the total salt content in the aboveground plant parts multiplied by the hay yield [77].

### 4.5. Comprehensive Evaluation of Oat Variety Production Performance and Comprehensive Evaluation of Soil Physicochemical Properties

The membership function method was applied to evaluate the production performance of these 15 oat varieties comprehensively. Additionally, the fuzzy mathematics membership function method was utilized to assess the soil physicochemical properties of different oat varieties. The membership function values for positively correlated indicators were calculated using Formula (8), while those for negatively correlated indicators were calculated using Formula (9).

#### 4.5.1. Membership Function Value of Comprehensive Indicators for Wheat Variety Production Performance


(5)
$$U(X_{j})=(X_{j}-X_{min})/(X_{max}-X_{min})\;j=1,2,3,\cdots ,n$$


In the formula, $X_{j}$ represents the j comprehensive index; $X_{min}$ represents the minimum value of the j comprehensive index; and $X_{max}$ represents the maximum value of the j comprehensive index.

#### 4.5.2. Comprehensive Indicator Weights for Oat Variety Production Performance


(6)
$$W_{j}=P_{j}/\sum_{j=1}^{n}P_{j}\;j=1,2,3,\cdots ,n$$


In the formula, $W_{j}$ represents the importance or weight of the j comprehensive indicator among all comprehensive indicators; $P_{j}$ represents the contribution rate of the comprehensive indicators of each part of oats obtained through principal component analysis.

#### 4.5.3. Comprehensive Evaluation of Production Performance in Oat Varieties


(7)
$$D=\sum_{j=1}^{n}\left[U(X_{j})\times W_{j}\right]\;j=1,2,3,\cdots ,n$$


In the formula, D represents the comprehensive evaluation value of the production performance of each oat variety.

#### 4.5.4. Comprehensive Evaluation of Soil Indicators for Oat Varieties


(8)
$$U(X_{j})=(X_{j}-X_{min})/(X_{max}-X_{min})\;j=1,2,3,\cdots ,n$$



(9)
$$U(X_{j})=1-(X_{j}-X_{min})/(X_{max}-X_{min})\;j=1,2,3,\cdots ,n$$


In the formula, $X_{j}$ represents the j comprehensive index; $X_{min}$ represents the minimum value of the j comprehensive index; and $X_{max}$ represents the maximum value of the j comprehensive index.

### 4.6. Data Processing and Analysis

In this study, principal component analysis was employed to reduce dimensionality and identify comprehensive indicators from 13 major performance metrics of 15 oat varieties at harvest. We utilized Microsoft Excel 2016 for data organization and employed DPS 7.05 and SPSS Statistics 22.0 for conducting single-factor difference significance analysis, principal component analysis, and membership function analysis. Additionally, GraphPad Prism 10.0 was utilized for graphic plotting, ensuring that our data visualization meets high academic standards.

## 5. Conclusions

The experimental results indicate significant differences in production performance and soil physicochemical properties among various oat varieties cultivated in the saline-alkali land of the Qaidam Basin. The comprehensive ranking of oat production performance at harvest is as follows: Qingtian No. 1 > Qingyin No. 1 > Molasses > Qingyin No. 2 > Xibei No. 1 > Jiayan No. 2 > Baler > Qinghai 444 > Qinghai sweet oat > Baiyan No. 7 > Haymaker > Mengshi No. 1 > Dafuweng > Lena > Gaoyan No. 1. Similarly, the comprehensive ranking of soil physicochemical properties at harvest is Qingtian No. 1 > Qingyin No. 1 > Molasses > Qinghai sweet oat > Dafuweng > Qingyin No. 2 > Baler > Qinghai 444 > Baiyan No. 7 > Mengshi No. 1 > Jiayan No. 2 > Haymaker > Lena > Xibei No. 1 > Gaoyan No. 1. These results demonstrate that the three oat varieties, Qingtian No. 1, Qingyin No. 1, and Molasses, consistently rank at the top for both production performance and soil physicochemical properties, indicating their superior suitability for cultivation in the saline-alkali lands of the Qaidam Basin.

## Figures and Tables

**Figure 1 plants-14-01978-f001:**
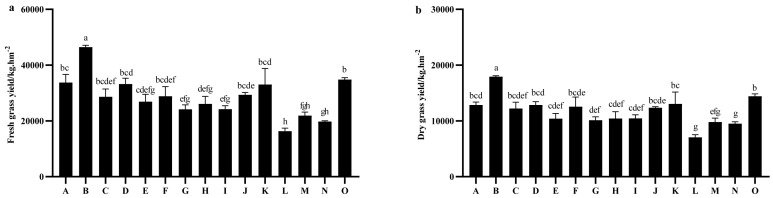
Comparison of fresh and hay yields of different oat (*Avena sativa* L.) varieties. (**a**) Fresh grass yield. (**b**) Dry grass yield. Note: A to O represents Qingtian No. 1, Molasses, Qinghai sweet oat, Qingyin No. 1, Qingyin No. 2, Dafuweng, Baiyan No. 7, Northwestern No. 1, Lena, Jiayan No. 2, Mengshi No. 1, Qinghai 444, Haymaker, Gaoyan No. 1, Baler, respectively. Different lowercase letters indicate significant differences among different breeds within the same growth period (*p* < 0.05).

**Figure 2 plants-14-01978-f002:**
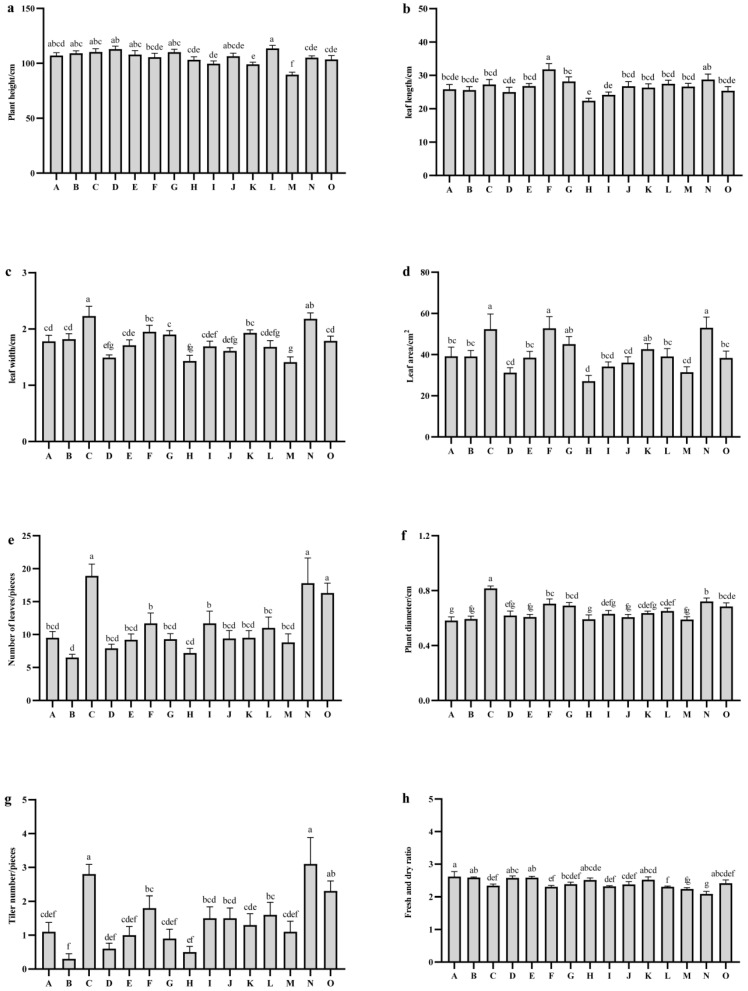
Comparison of agronomic traits of different oat varieties. (**a**) Plant height. (**b**) leaf length. (**c**) leaf width. (**d**) Leaf area. (**e**) Number of leaves. (**f**) Plant diameter. (**g**) Tiler number. (**h**) Fresh and dry ratio. Note: A to O represents Qingtian No. 1, Molasses, Qinghai sweet oat, Qingyin No. 1, Qingyin No. 2, Dafuweng, Baiyan No. 7, Northwestern No. 1, Lena, Jiayan No. 2, Mengshi No. 1, Qinghai 444, Haymaker, Gaoyan No. 1, Baler, respectively. Different lowercase letters indicate significant differences among different breeds within the same growth period (*p* < 0.05).

**Figure 3 plants-14-01978-f003:**
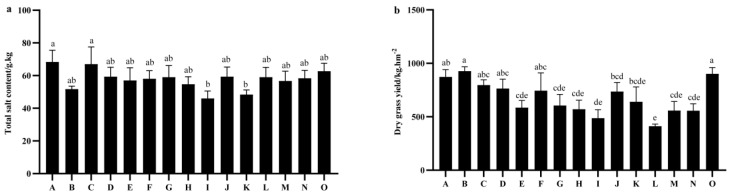
Comparison of total salt content and salt accumulation in different oat varieties. (**a**) Total salt content. (**b**) Dry grass yield. Note: A to O represents Qingtian No. 1, Molasses, Qinghai sweet oat, Qingyin No. 1, Qingyin No. 2, Dafuweng, Baiyan No. 7, Northwestern No. 1, Lena, Jiayan No. 2, Mengshi No. 1, Qinghai 444, Haymaker, Gaoyan No. 1, Baler, respectively. Different lowercase letters indicate significant differences among different breeds within the same growth period (*p* < 0.05).

**Figure 4 plants-14-01978-f004:**
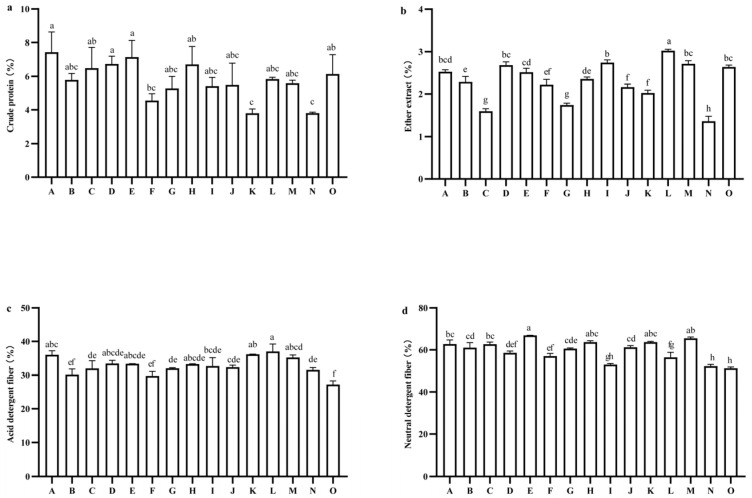
Comparison of nutritional quality of different oat varieties. (**a**) Crude protein. (**b)** Ether extract. (**c**) Acid detergent fiber. (**d**) Neutral detergent fiber. Note: A to O represent Qingtian No. 1, Molasses, Qinghai sweet oat, Qingyin No. 1, Qingyin No. 2, Dafuweng, Baiyan No. 7, Northwestern No. 1, Lena, Jiayan No. 2, Mengshi No. 1, Qinghai 444, Haymaker, Gaoyan No. 1, Baler, respectively. Different lowercase letters indicate significant differences among different breeds within the same growth period (*p* < 0.05).

**Figure 5 plants-14-01978-f005:**
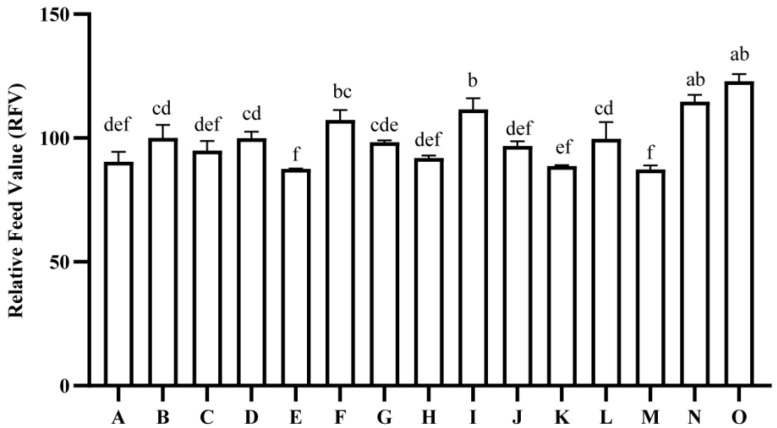
Comparison of relative feeding value of different oat varieties. Note: A to O represent Qingtian No. 1, Molasses, Qinghai sweet oat, Qingyin No. 1, Qingyin No. 2, Dafuweng, Baiyan No. 7, Northwestern No. 1, Lena, Jiayan No. 2, Mengshi No. 1, Qinghai No. 444, Haymaker, Gaoyan No. 1, Baler, respectively. Different lowercase letters indicate significant differences among different breeds within the same growth period (*p* < 0.05).

**Figure 6 plants-14-01978-f006:**
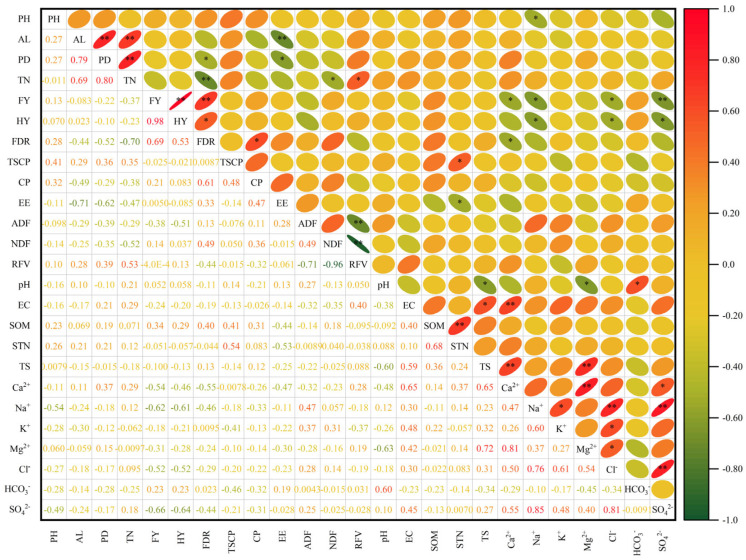
Correlation analysis of cultivated oat production performance and soil physical and chemical properties. Note: * was significantly correlated at the 0.05 level, and ** was significantly correlated at the 0.01 level. PH is plant height; AL is flag leaf; PD is stem diameter; TN is the number of tillers; FY is fresh grass yield; HY is hay yield; FDR is the ratio of fresh to dry; TSCP is the total salt content in plants; CP is crude protein; EE is crude fat; ADF is acid detergent fiber; NDF is neutral detergent fiber; RFV is relative feeding value; EC is soil electrical conductivity; SOM is soil organic matter; STN is soil total nitrogen; TS is soil total salt.

**Table 1 plants-14-01978-t001:** Comparison of the effects of different oat varieties on soil pH.

Variety	Seedling Stage 0–15 cm	Harvest Period 0–15 cm	Seedling Stage 15–30 cm	Harvest Period 15–30 cm
CK	8.81 ± 0.18 a	8.92 ± 0.22 a	8.91 ± 0.08 a	9.16 ± 0.15 a
Qingtian No. 1	8.72 ± 0.17 abc	8.50 ± 0.07 b	8.66 ± 0.18 abcde	8.63 ± 0.18 bc
Molasses	8.52 ± 0.08 bcde	8.47 ± 0.11 bc	8.85 ± 0.13 ab	8.44 ± 0.11 cd
Qinghai sweet oat	8.30 ± 0.07 e	8.14 ± 0.16 de	8.73 ± 0.08 abcd	8.03 ± 0.04 ef
Qingyin No. 1	8.46 ± 0.14 cde	8.17 ± 0.21 de	8.72 ± 0.08 abcd	8.16 ± 0.14 de
Qingyin No. 2	8.32 ± 0.11 de	8.24 ± 0.03 cd	8.75 ± 0.18 abc	8.04 ± 0.10 ef
Dafuweng	8.64 ± 0.08 abc	8.07 ± 0.06 def	8.82 ± 0.18 ab	7.95 ± 0.06 ef
Baiyan No. 7	8.62 ± 0.07 abc	8.05 ± 0.11 def	8.55 ± 0.13 cde	8.00 ± 0.23 ef
Northwestern No. 1	8.68 ± 0.07 abc	7.88 ± 0.08 f	8.44 ± 0.08 e	7.83 ± 0.08 f
Lena	8.70 ± 0.01 abc	7.99 ± 0.10 ef	8.83 ± 0.10 ab	7.90 ± 0.12 ef
Jiayan No. 2	8.48 ± 0.19 cde	8.18 ± 0.08 de	8.48 ± 0.12 de	8.09 ± 0.11 ef
Mengshi No. 1	8.59 ± 0.11 abcd	8.54 ± 0.03 b	8.71 ± 0.10 abcde	8.68 ± 0.16 bc
Qinghai No. 444	8.72 ± 0.12 abc	8.52 ± 0.10 b	8.79 ± 0.18 abc	8.80 ± 0.22 b
Haymaker	8.58 ± 0.11 abcd	8.5 ± 0.09 b	8.72 ± 0.12 abcd	8.67 ± 0.22 bc
Gaoyan No. 1	8.78 ± 0.19 ab	8.57 ± 0.02 b	8.63 ± 0.13 bcde	8.62 ± 0.04 bc
Baler	8.81 ± 0.19 a	8.60 ± 0.15 b	8.83 ± 0.14 ab	8.54 ± 0.22 bc
F Value	2.87 **	11.27 **	2.09 *	13.11 **

Note: CK is saline-alkali open space control. The values in the table are mean ± standard error; different letters indicate significant differences among different breeds in the same column (*p* < 0.05). ** indicates highly significant differences at the 0.01 level; * indicates significant differences at the 0.05 level.

**Table 2 plants-14-01978-t002:** Comparison of effects of different oat varieties on soil electrical conductivity.

Variety	Seedling Stage 0–15 cm Soil EC ms/m	Harvest Period 0–15 cm Soil Conductivity	Seedling Stage 15–30 cm Soil Conductivity	Harvest Period 15–30 cm Soil EC ms/m
CK	0.70 ± 0.08 a	0.77 ± 0.05 a	0.80 ± 0.08 a	0.83 ± 0.05 a
Qingtian No. 1	0.40 ± 0 de	0.20 ± 0 fg	0.37 ± 0.05 fgh	0.17 ± 0.05 fg
Molasses	0.23 ± 0.05 f	0.20 ± 0 fg	0.70 ± 0 ab	0.13 ± 0.05 g
Qinghai sweet oat	0.47 ± 0.05 cd	0.43 ± 0.09 bc	0.47 ± 0.05 def	0.17 ± 0.05 fg
Qingyin No. 1	0.37 ± 0.05 de	0.33 ± 0.05 de	0.27 ± 0.05 h	0.27 ± 0.05 cdef
Qingyin No. 2	0.53 ± 0.09 bc	0.20 ± 0 fg	0.33 ± 0.05 gh	0.33 ± 0.05 cd
Dafuweng	0.57 ± 0.05 bc	0.20 ± 0 fg	0.50 ± 0.08 cde	0.30 ± 0 cde
Baiyan No. 7	0.63 ± 0.05 ab	0.27 ± 0.05 ef	0.77 ± 0.05 a	0.20 ± 0 efg
Northwestern No. 1	0.47 ± 0.05 cd	0.47 ± 0.05 b	0.53 ± 0.05 cd	0.47 ± 0.05 b
Lena	0.47 ± 0.09 cd	0.37 ± 0.05 cd	0.33 ± 0.05 gh	0.33 ± 0.05 cd
Jiayan No. 2	0.53 ± 0.05 bc	0.17 ± 0.05 g	0.53 ± 0.09 cd	0.37 ± 0.05 bc
Mengshi No. 1	0.33 ± 0.05 ef	0.27 ± 0.05 ef	0.43 ± 0.09 defg	0.27 ± 0.05 cdef
Qinghai No. 444	0.40 ± 0.08 de	0.20 ± 0 fg	0.33 ± 0.05 gh	0.23 ± 0.05 defg
Haymaker	0.40 ± 0.08 de	0.20 ± 0 fg	0.47 ± 0.05 def	0.23 ± 0.05 defg
Gaoyan No. 1	0.37 ± 0.05 de	0.37 ± 0.05 cd	0.40 ± 0.08 efg	0.37 ± 0.12 bc
Baler	0.37 ± 0.05 de	0.30 ± 0 de	0.60 ± 0 bc	0.47 ± 0.05 b
F Value	7.66 **	30.93 **	13.72 **	20.51 **

Note: CK is saline-alkali open space control. The values in the table are mean ± standard error; different letters indicate significant differences among different breeds in the same column (*p* < 0.05). ** indicates highly significant differences at the 0.01 level.

**Table 3 plants-14-01978-t003:** Comparison of effects of different oat varieties on soil total salt content.

Variety	Seedling Stage 0–15 cm Total Salt Content	Harvest Period 0–15 cm Total Salt Content	Seedling Stage 15–30 cm Total Salt Content	Harvest Period 15–30 cm Total Salt Content
CK	2.45 ± 0.06 a	2.53 ± 0.17 a	2.38 ± 0.03 a	2.44 ± 0.05 a
Qingtian No. 1	1.06 ± 0.06 g	0.9 ± 0.09 cd	0.75 ± 0.04 i	0.72 ± 0.09 fg
Molasses	0.61 ± 0.07 h	0.51 ± 0.03 e	0.99 ± 0.06 h	0.69 ± 0.06 fg
Qinghai sweet oat	1.44 ± 0.07 f	1.05 ± 0.05 c	1.49 ± 0.04 e	0.31 ± 0.07 i
Qingyin No. 1	1.34 ± 0.09 f	0.91 ± 0.07 cd	1.16 ± 0.09 fg	0.31 ± 0.07 i
Qingyin No. 2	1.8 ± 0.08 d	0.88 ± 0.06 d	1.26 ± 0.09 f	0.87 ± 0.07 de
Dafuweng	1.68 ± 0.03 de	0.76 ± 0.01 d	1.84 ± 0.05 d	0.6 ± 0.04 g
Baiyan No. 7	2.06 ± 0.08 c	1.26 ± 0.08 b	1.87 ± 0.07 d	1.49 ± 0.05 c
Northwestern No. 1	1.58 ± 0.05 e	1.36 ± 0.05 b	1.81 ± 0.05 d	1.73 ± 0.04 b
Lena	1.33 ± 0.05 f	1.31 ± 0.08 b	1.13 ± 0.06 g	0.85 ± 0.03 de
Jiayan No. 2	1.78 ± 0.08 d	0.41 ± 0.02 e	1.91 ± 0.03 cd	0.96 ± 0.05 d
Mengshi No. 1	2.06 ± 0.02 c	0.77 ± 0.04 d	2.02 ± 0.04 c	0.61 ± 0.06 g
Qinghai No. 444	1.72 ± 0.06 d	0.38 ± 0.07 e	0.97 ± 0.06 h	0.46 ± 0.08 h
Haymaker	2.27 ± 0.02 b	0.36 ± 0.07 e	1.57 ± 0.04 e	0.46 ± 0.02 h
Gaoyan No. 1	1.37 ± 0.03 f	0.88 ± 0.09 d	2.18 ± 0.09 b	0.79 ± 0.06 ef
Baler	1.01 ± 0.07 g	0.87 ± 0.06 d	1.24 ± 0.02 fg	0.9 ± 0.02 de
F Value	119.95 **	91.82 **	147.09 **	182.29 **

Note: CK is saline-alkali open space control. The values in the table are mean ± standard error; different letters indicate significant differences among different breeds in the same column (*p* < 0.05). ** indicates highly significant differences at the 0.01 level.

**Table 4 plants-14-01978-t004:** Characteristic vector and contribution rate of principal component analysis of 15 oat varieties.

Index	Principal Component 1	Principal Component 2	Principal Component 3	Principal Component 4
Plant height	−0.035	0.240	0.373	0.293
Flag leaf area	−0.358	0.106	0.247	−0.253
Stem thickness	−0.384	0.079	0.257	−0.043
Tillering number	−0.410	−0.039	0.105	0.056
Fresh grass yield	0.166	0.528	−0.081	−0.192
Hay yield	0.091	0.540	−0.125	−0.219
Fresh-to-dry ratio	0.364	0.293	0.118	−0.005
Total salt content	−0.083	0.126	0.528	0.289
Crude Protein	0.271	0.127	0.310	0.420
Crude Fat	0.270	−0.094	−0.228	0.477
Acid Detergent Fiber	0.212	−0.417	0.194	−0.073
Neutral Detergent Fiber	0.304	−0.098	0.331	−0.389
Relative Feeding Value.	−0.311	0.205	−0.337	0.341
Eigenvalue	4.744	2.742	2.036	1.598
Contribution rate/%	36.496	21.094	15.664	12.290
Cumulative contribution rate/%	36.496	57.590	73.254	85.544

**Table 5 plants-14-01978-t005:** Comprehensive index value, weight, membership function value, *D* value, and comprehensive evaluation of 15 oats.

Variety	Principal Component				Subordinate Function Values				*D* Value
	1	2	3	4	*U*(X_1_)	*U*(X_2_)	*U*(X_3_)	*U*(X_4_)	
Qingtian No. 1	2.359	0.627	1.829	0.646	0.998	0.562	0.784	0.702	0.809
Molasses	1.918	3.359	−1.197	−1.151	0.938	1.000	0.266	0.345	0.745
Qinghai sweet oat	−2.552	0.873	3.095	−0.588	0.328	0.601	1.000	0.457	0.537
Qingyin No. 1	1.885	1.084	0.167	1.308	0.933	0.635	0.500	0.834	0.766
Qingyin No. 2	2.147	−0.367	1.244	0.067	0.969	0.402	0.684	0.587	0.722
Dafuweng	−2.269	0.794	−0.409	−0.460	0.367	0.589	0.401	0.482	0.444
Baiyan No. 7	−0.909	−0.224	0.928	−0.519	0.552	0.425	0.630	0.470	0.523
Northwestern No. 1	2.376	−0.801	−0.202	0.148	1.000	0.333	0.437	0.603	0.675
Lena	−0.458	−1.075	−2.756	0.833	0.613	0.289	0.000	0.740	0.439
Jiayan No. 2	0.281	0.086	0.011	−0.236	0.714	0.475	0.473	0.527	0.584
Mengshi No. 1	0.759	−0.597	−0.770	−2.880	0.779	0.366	0.339	0.000	0.485
Qinghai No. 444	−0.252	−2.485	0.647	2.140	0.642	0.063	0.582	1.000	0.539
Haymaker	1.470	−2.876	−0.847	−0.741	0.876	0.000	0.326	0.426	0.495
Gaoyan No. 1	−4.957	−0.941	−0.285	−0.481	0.000	0.310	0.422	0.478	0.223
Baler	−1.799	2.544	−1.454	1.914	0.431	0.869	0.223	0.955	0.576
Weight					0.427	0.247	0.183	0.1434	

## Data Availability

The original contributions presented in this study are included in the article. Further inquiries can be directed to the corresponding author.

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
