# Peer review of "Comparative Study on Production Performance of Different Oat (Avena sativa) Varieties and Soil Physicochemical Properties in Qaidam Basin"

_plants, 2025, doi:10.3390/plants14131978_

Round 1

Reviewer 1 Report

Comments and Suggestions for Authors

Soil salinity is among the important abiotic stress conditions that threaten plant production today. Therefore, it causes both plant and economic product losses. The authors have worked on an important problem and it is an original study. The oat plant is discussed in the article and the references used are relevant to the subject. Important outputs have been provided in the article written in a good language. However, some points need to be improved.

The title of the article seems very confusing. Simplify it and add the Latin name of the oat plant.

I recommend that developments such as background, material and method be removed from the abstract.

When I examine the introduction section, I recommend that information about the species studied should be given first, then soil salinity should be passed on and the purpose of the study should be finalized.

I recommend that the negative effects of soil salinity be written in more detail physiologically in the introduction section (cell damage etc.)

Information such as oat production value, important producer countries etc. should be added.

Varieties are mentioned in line 59. It would be better to use the word "cultivar" in the culture varieties.

In the material and method section, it should be written in which year the study was conducted. Also, meteorological data can be presented in a table or a simple graph (just a suggestion)

Please use "cultivar" in the test material section.

Can you explain why these varieties were chosen? What are their features in a few sentences.

How was the 30 square meters in line 620 determined?

No references were added to some long sentences in the method section. Please add a reference to them.

The PCA analysis in line 686 should be written under the data analysis heading.

There are many formulas in the study. Can it be given as an additional table?

When I examine the findings and discussion section of the study, it is well prepared and discussed with sufficient references. However, the graphs can be prepared better visually.

Also, I see that there are many tables. It would be better to put them in a graph and give the tables as supplementary.

Also, figures 1,2,3,4,5 are not readable. Image quality needs to be improved.

Why was no correlation analysis performed in such a comprehensive and beautiful study?

In the Discussion section, it is stated that soil salinity is an important abiotic parameter in affecting yield, but it would be better to add information about other parameters.

PCA analysis graphics need to be added to the study.

My general comment;
Important outputs were obtained in the study. Instead of using too many tables in the article written in a beautiful language, presenting it with colorful and flashy graphics will add modernity to the study. I believe that the article can be accepted after the above suggestions are made.

Author Response

Point-by-point response to the reviewer comments

Dear Reviewers,

Thank you for your thoughtful suggestions and insights, which have benefited from the manuscript. I am looking forward to working with you to move this manuscript closer to publication in “Plants”.

The manuscript has been rechecked and the necessary changes have been made in accordance with your suggestions. The responses to all comments have been prepared and attached below. We have tried our best to solve the problems you proposed, and we hope that the revised manuscript is now suitable for publication in the journal “Plants”. If you have any questions remained about this paper, please feel free to contact us.

Reviwer #1:

The title of the article seems very confusing. Simplify it and add the Latin name of the oat plant.

Re: Thank you very much for your suggestions. We have made the modifications. Thank you. (line 2-line 4)

I recommend that developments such as background, material and method be removed from the abstract.

Re: Thank you for your suggestions. We have modified the content related to the abstract. Thank you. (line 10-line 24)

When I examine the introduction section, I recommend that information about the species studied should be given first, then soil salinity should be passed on and the purpose of the study should be finalized.

Re: Thank you very much for your review. Because of the large area of Qaidam Basin and the wide area of saline-alkali land in this study, the degree of saline-alkali in different regions varies greatly. By screening oat varieties suitable for local planting, it provides scientific basis and theoretical reference for agricultural and animal husbandry production. It has reference value for adopting the ' planting suitable land ' model to control saline-alkali land. Therefore, we think it is more appropriate to introduce saline-alkali land first.

I recommend that the negative effects of soil salinity be written in more detail physiologically in the introduction section (cell damage etc.)

Re: We greatly appreciate your review. In accordance with your suggestions, we have added the relevant content to the manuscript. Please kindly review it. (line 53)

Information such as oat production value, important producer countries etc. should be added.

Re: Thank you very much for your valuable comments on this study, we have added the value of oat production and the main producing countries in the introduction. (line 69-line 73)

Varieties are mentioned in line 59. It would be better to use the word "cultivar" in the culture varieties.

Re: Thank you very much for your review, we have been modified. (line 82)

In the material and method section, it should be written in which year the study was conducted. Also, meteorological data can be presented in a table or a simple graph (just a suggestion)

Re: Thank you very much for your suggestion, we have indicated in the experimental design and method that the study will take place in 2023 (line 735), and the basic meteorological data are described in the test site profile. (line 714-line 719)

Please use "cultivar" in the test material section.

Re: Thank you very much for your review, we have been modified. (line 725)

Can you explain why these varieties were chosen? What are their features in a few sentences.

Re: Thank you very much for your advice. For your questions, our answers are as follows. There is a large area of saline-alkali land in Qinghai Province. Before this experiment, we selected some oat varieties with wide planting and strong adaptability in Qinghai Province. In the early stage, different concentrations of salt stress treatment were carried out in the laboratory, and 15 oat varieties with high salt tolerance were selected for field test. The production performance of these 15 oat varieties and the changes of soil physical and chemical properties were compared through field test. The field test was more suitable for agricultural and animal husbandry production, providing scientific basis and theoretical reference for agricultural and animal husbandry production.

How was the 30 square meters in line 620 determined?

Re: Thank you very much for your review. The reason why we have a plot area of 30 square meters is that the large plot area has high spatial variability and may require a larger area to cover typical microenvironment differences. Statistical calculations ( such as analysis of variance ) may indicate that the area of 30 square meters can reach the 95 % confidence level and the efficacy of 0.8, balancing the needs of data reliability and experimental efficiency. In the comparison experiment of crop varieties, 30 square meters of plot can reduce the influence of edge effect and ensure that the yield data is statistically significant.

No references were added to some long sentences in the method section. Please add a reference to them.

Re: We have added corresponding references to the relevant content. (line 754) (line 765)(line 772)

The PCA analysis in line 686 should be written under the data analysis heading.

Re: Thank you very much for your review, we have revised the corresponding content. (line 828-line 830)

There are many formulas in the study. Can it be given as an additional table?

Re: Thank you very much for your review. For the problem that your proposed formula is presented in the form of a schedule. The journal template is required to be added to the text, so we put it in the text.

When I examine the findings and discussion section of the study, it is well prepared and discussed with sufficient references. However, the graphs can be prepared better visually.

Re: Thank you very much for this very good suggestion, I think it is very critical. In this paper, the data of the above-ground part of the cultivated oat varieties on the production performance are all presented in the form of drawing, but the soil physical and chemical properties are partly due to the fact that there are 15 cultivated oat varieties in this experiment, which are divided into 2 periods and 2 soil layers. If it is presented in the form of histogram, it will look very dense and the letter mark will be crowded. Therefore, we think that it will be more convenient for readers to read it in the form of tables by comparison.

Also, I see that there are many tables. It would be better to put them in a graph and give the tables as supplementary.

Re: Thank you for your review. The tables on soil physical and chemical properties in this paper are all presented in the form of schedules. It is named in turn by Table S1, Table S2, and Table S3. Putting the tables together will make it difficult for readers to read, and readers cannot locate a certain soil physical and chemical properties index faster according to the reading content. Named in turn with Table S1, Table S2, and Table S3, it can be positioned to a certain indicator that is needed more quickly, making reading easier.

Also, figures 1,2,3,4,5 are not readable. Image quality needs to be improved.

Re: Thank you for your review. We have modified the relevant pictures in the text and uploaded all the pictures in the attachment, please review.

Why was no correlation analysis performed in such a comprehensive and beautiful study?

Re: Thank you for your review. We have conducted a correlation analysis in the article, please review. (line 500-line 523)

In the Discussion section, it is stated that soil salinity is an important abiotic parameter in affecting yield, but it would be better to add information about other parameters.

Re: Thank you for your review. It has been mentioned in the introduction that soil salinization occurs in arid and semi-arid areas (line 31-line 32)(line 43-line 45)(line 56-line 57). Saline-alkali is accompanied by drought. Soil salinization destroys soil aggregate structure, accelerates soil compaction, and reduces cultivable area. Qaidam area is an extremely arid area with low annual precipitation and high salinity. The problem of soil salinization is becoming more and more serious, which has affected about 42.5 % of the agricultural land in the basin.

PCA analysis graphics need to be added to the study.

Re: Thank you for your review. In this experiment, the principal component analysis combined with membership function was used to comprehensively evaluate the production performance of oats. The PCA diagram and the following membership function analysis could not be better connected. The combination of the two methods was the best combination. The feature vector data can reflect the contribution of a certain index to this principal component. Principal component analysis and membership function method are both methods for comprehensive evaluation of plant indicators. Principal component analysis can be used alone for evaluation, but principal component analysis needs to transform the original data. Some studies have shown that the data transformed by membership function is more reasonable than other methods. The membership function method can also be used alone for evaluation, but the calculation is too cumbersome when there are too many evaluation indicators. At present, most of the research is the combination of principal component analysis and membership function method.

Reference:

Guan Hao, Xu Duo, Li Haiping, et al. Study on nutritional quality and rumen degradation characteristics of 17 oat varieties in alpine regions [J]. Acta Prataculturae Sinica, 2024, 33(09): 185-198.

Wang Miaomiao, Zhou Xiangrui, Liang Guoling, et al. Comprehensive evaluation of salt tolerance at seedling stage of 5 oat materials [J]. Acta Prataculturae Sinica, 2020, 29(08): 143-154.

Meng Chen, Lu Xueli, Song Yiru, et al. Evaluation and identification of salt tolerance at seedling stage of 11 Leonurus japonicus germplasm materials [J]. Acta Prataculturae Sinica, 2024, 33(05): 196-203.

Reviewer 2 Report

Comments and Suggestions for Authors

Dear Authors,

Please find my recommendation for "Comparative Study on Production Performance of Different Oat Varieties and Soil Physicochemical Properties in Saline-Alkali Land of Qaidam Basin":

L31-33: “Soil salinization has emerged as a critical agricultural challenge that significantly hampers agricultural development” is a vague statement in my opinion. I recommend for authors to provide specific data on economic losses or food security in order to strengthen the justification for the study

L33: While the global extent of saline-alkali land is cited (around 800 million hectares), this statistic is not critically analyzed or contextualized within the study area, missing an opportunity to justify the regional focus

L36-51: Although the authors mention the effects of soil salinity on crop growth and soil structure, there is a lack connection to the unique challenges in Qinghai or the Qaidam Basin. Please elaborate it to emphasize the regional relevance

L45: Please replace "6]," with "6]"

L45-46: The rationale for evaluating salt-tolerant forage germplasm resources is reasonable, but in my opinion the introduction does not clearly state the gap in current knowledge or why existing research is insufficient. Please consider this in a deeper manner

L53-55: The focus on oats is justified by their salt tolerance and nutritional value, but in my view  the introduction does not sufficiently connect this to the specific needs or constraints of the Qaidam Basin’s saline-alkali land

L58-61: The discussion of plant adaptive mechanisms to salt stress is broad and generic; integrating more region- or crop-specific references would improve the research relevance

L68-74: Please better formulate the research hypothesis or specific research questions

General recommendation:

Introduction - Statements about the impact of soil salinization on soil structure and crop yield are accurate but would be more compelling if supported by specific regional examples or references. Similarly, statements about the negative effects of saline-alkali stress on oats would be more persuasive if supported by prior studies or region-specific data.

L601: Please present soil characteristics details from the initials of the experiment, such as initial soil salinity, texture, and baseline nutrient levels, etc. It is important to understand the starting conditions

L611:  Please mention it they were chosen based on prior salt tolerance data, genetic diversity, or other criteria. This need clarification

L619-620: The experimental design states a "randomized block design" with plots of 30 m² (5 m × 6 m) and three replications, but it does not specify how blocks were arranged, whether blocks were spatially separated to minimize edge effects, or how randomization was performed. Please detail this

L623: The sowing date (June 13, 2023) is provided, but there is no mention of whether planting was synchronized across all plots or if any measures were taken to ensure uniform seed distribution and depth

General recommendation: In the manuscript the authors refer to "saline-alkali open space control," but it is not clear what is beyond this, if they are non-saline or non-alkali soil controls. Without clear information in this sense is hard to consider all baseline comparisons

L77: Yield and trait data are presented with means and significance indicators, but the specific statistical tests used are not clearly specified. Please do that and consider this during all results section where is the case

General comment: The results in my opinion are densely presented which may overwhelm readers, I recommend their reconsideration and those extensive or less relevant to be moved at supplementary materials. Please increase the font sizes from insides of the graphs

L508: The discussion claims that oats are "recognized as a moderately salt-tolerant crop and can be utilized to enhance saline-alkali land" but for me there are not enough referencing specific prior studies or providing a comprehensive comparison of salt tolerance levels among different crops, that could strengthen the claim

L509: The statement that "the successful introduction of oats is primarily determined by the completeness of their growth and development" is in my opinion is very simplistic and lacks depth; it overlooks other critical factors such as genetic variability, soil conditions, management practices, and environmental stresses. This should be considered

L519: The comparison of yield reductions under saline-alkali conditions with previous studies (e.g., Bale Fresh Grass yield) is somewhat superficial as it does not account for differences in experimental conditions, soil salinity levels, or management practices, which limits the validity of the direct comparison

L527: The assertion that "the salt accumulation in Mulesi reached 927.13 kg·hm-2 under identical conditions, surpassing Qinghai 444 by 515.10 kg·hm-2" implies a positive correlation between salt accumulation and yield without discussing potential thresholds or the possibility that excessive salt accumulation could be detrimental in my opinion

L533: please replace "li et al., " with "Li et al., "

L541-558: In my opinion the explanation for lower crude protein and RFV values in this study compared to previous literature is somewhat speculative. It attributes the decline mainly to "adverse effects of saline-alkali soil" without providing direct evidence or discussing possible methodological differences

L551: check "li et al.[22]."

L560: The discussion on soil physicochemical properties notes a decrease in pH, EC, and total salt content over the growth period, suggesting these are beneficial effects. However, it does not sufficiently address whether these reductions are sustainable or whether they could be transient or due to experimental artifacts

L564: In my opinion the claim that "the reduction in soil salinity was greater in the 15-30 cm soil layer" is made without discussing the mechanisms behind this pattern, such as leaching, plant uptake, or soil processes, which would add depth to the interpretation

L569: The statement that "the soil EC mostly exceeded 4 dS/m, categorizing them as saline-alkali land" is presented as a fact, but it lacks discussion on the implications for crop growth, thresholds for different crops, or how these levels compare to known salinity tolerance limits

L572: The discussion suggests that "the primary mechanism of salt tolerance might be osmotic stress tolerance" based on observed reductions in soil salt content, but it does not critically evaluate alternative mechanisms such as ion exclusion or tissue tolerance, nor does it cite relevant mechanistic studies

L587: The discussion on K+ and Na+ levels attributes their patterns to "competitive inhibition" without referencing specific studies or mechanisms, and it oversimplifies the complex ion uptake and transport processes involved in salt stress responses

L594: The explanation of HCO3- reduction and its role in plant growth is brief and lacks detailed mechanistic insight or references to relevant physiological studies in my opinion

General comments: In my opinion to build all the discussion on comparison/correlating the obtained research data with a single article "Li et al.," is very scarce diminishing seriously the scientific relevance. On other way, the discussion tends to be descriptive rather than critically analytical, often attributing observed phenomena to general concepts without sufficient mechanistic or literature support, reducing the scientific rigor of the interpretation in my view.

Author Response

Point-by-point response to the reviewer comments

Dear Reviewers,

Thank you for your thoughtful suggestions and insights, which have benefited from the manuscript. I am looking forward to working with you to move this manuscript closer to publication in “Plants”.

The manuscript has been rechecked and the necessary changes have been made in accordance with your suggestions. The responses to all comments have been prepared and attached below. We have tried our best to solve the problems you proposed, and we hope that the revised manuscript is now suitable for publication in the journal “Plants”. If you have any questions remained about this paper, please feel free to contact us.

Reviwer #2:

L31-33: “Soil salinization has emerged as a critical agricultural challenge that significantly hampers agricultural development” is a vague statement in my opinion. I recommend for authors to provide specific data on economic losses or food security in order to strengthen the justification for the study

Re: Thank you for your review. We have added relevant content in the introduction. (line 34-line 36)

L33: While the global extent of saline-alkali land is cited (around 800 million hectares), this statistic is not critically analyzed or contextualized within the study area, missing an opportunity to justify the regional focus

Re: Thank you very much for your valuable comments. We have added relevant reference support on the study area in this part. (line 39-line 42)(line 55-line 58)

L36-51: Although the authors mention the effects of soil salinity on crop growth and soil structure, there is a lack connection to the unique challenges in Qinghai or the Qaidam Basin. Please elaborate it to emphasize the regional relevance

Re: Thank you very much for your valuable suggestions. This part has been supported by relevant references on the increasingly serious soil salinization in the Qaidam Basin. (line 55-line 58)

L45: Please replace "6]," with "6]"

Re: Thank you very much for your review. We have made revisions to the manuscript. (line 61)

L45-46: The rationale for evaluating salt-tolerant forage germplasm resources is reasonable, but in my opinion the introduction does not clearly state the gap in current knowledge or why existing research is insufficient. Please consider this in a deeper manner

Re: We have added the shortcomings of the existing research in the introduction, please review it. (line 93-line 102)

L53-55: The focus on oats is justified by their salt tolerance and nutritional value, but in my view the introduction does not sufficiently connect this to the specific needs or constraints of the Qaidam Basin’s saline-alkali land

Re: We have added relevant content to the introduction of the manuscript. (line 58-line 61)(line 71-line 73)

L58-61: The discussion of plant adaptive mechanisms to salt stress is broad and generic; integrating more region- or crop-specific references would improve the research relevance

Re: We have added related research in the discussion section. (line 85-line 88)

L68-74: Please better formulate the research hypothesis or specific research questions

Re: We have reorganized the research hypothesis in the last part of the introduction. (line 103-line 115)

General recommendation:

Introduction - Statements about the impact of soil salinization on soil structure and crop yield are accurate but would be more compelling if supported by specific regional examples or references. Similarly, statements about the negative effects of saline-alkali stress on oats would be more persuasive if supported by prior studies or region-specific data.

Re: We have added previous studies in the introduction to prove that saline-alkali land does reduce oat yield and oat nutritional quality. (line 85-line 88)

L601: Please present soil characteristics details from the initials of the experiment, such as initial soil salinity, texture, and baseline nutrient levels, etc. It is important to understand the starting conditions

Re: We have added test geological and basic nutrient information to the test site profile. (line 721-line 723) Before the experiment, the soil salt content of 0-30cm soil layer was determined to be 2.40g·kg-1 and the soil pH was 8.46 by five-point sampling method. The physical and chemical properties of the soil were tested. In the article, soil samples were collected in each plot during the oat seedling stage and oat harvest period, and the physical and chemical properties of the soil were compared before and after. Therefore, the initial conditions such as salt content and pH of the entire plot were not described in the article.

L611:  Please mention it they were chosen based on prior salt tolerance data, genetic diversity, or other criteria. This need clarification

Re: Thank you very much for your advice. For your questions, our answers are as follows. There is a large area of saline-alkali land in Qinghai Province. Before this experiment, we selected some oat varieties with wide planting and strong adaptability in Qinghai Province. In the early stage, different concentrations of salt stress treatment were carried out in the laboratory, and 15 oat varieties with high salt tolerance were selected for field test. The production performance of these 15 oat varieties and the changes of soil physical and chemical properties were compared through field test. The field test was more suitable for agricultural and animal husbandry production, providing scientific basis and theoretical reference for agricultural and animal husbandry production.

L619-620: The experimental design states a "randomized block design" with plots of 30 m² (5 m × 6 m) and three replications, but it does not specify how blocks were arranged, whether blocks were spatially separated to minimize edge effects, or how randomization was performed. Please detail this

Re: This experiment was completely random. For example, after the first plot of Qingtian No.1 oat variety was planted in this plot, the second plot of Qingtian No.1 may be arranged behind other oat variety plots, so the three plots of each cultivated oat variety were completely random.

L623: The sowing date (June 13, 2023) is provided, but there is no mention of whether planting was synchronized across all plots or if any measures were taken to ensure uniform seed distribution and depth

Re: In the test material, we mentioned that the test was artificially furrowed and sown. It was manually controlled, with a plot interval of 0.5 m, a row spacing of 15 cm, and a sowing depth of 3 cm. The tested oats were uniformly sown on June 13,2023. The sowing date of all cultivated oat varieties was the same, and the sowing amount was 150 kg·hm-2, and then converted into the sowing amount of each row. The sowing amount of each row was weighed by an electronic balance and then artificially sown.

General recommendation: In the manuscript the authors refer to "saline-alkali open space control," but it is not clear what is beyond this, if they are non-saline or non-alkali soil controls. Without clear information in this sense is hard to consider all baseline comparisons

Re: In this paper, oat varieties collected in Qaidam Basin were used as experimental materials. Based on the analysis data of agronomic traits, quality and soil physical and chemical properties of different oat varieties, oat varieties suitable for planting in the study area were comprehensively evaluated and screened. Screening high-yield and high-quality forage varieties suitable for specific regional climate and soil conditions, optimizing the structure of forage industry, enriching forage supply, and improving the efficiency of animal husbandry production provide a basis for selecting oat varieties suitable for planting in the study area. This has important guiding significance for the promotion of oat planting to new regions or environments, and provides a scientific basis for local agricultural production. The area of Qaidam Basin is large, and the area of saline-alkali land is wide, and the degree of saline-alkali in different regions is different. By screening oat varieties suitable for local planting, it has reference value for adopting the mode of ' planting suitable land ' to control saline-alkali land. Therefore, we did not carry out the control treatment test of cultivated oat varieties in non-saline-alkali land, but we discussed a lot of the previous oat experiments of the same varieties in non-saline-alkali land, and discussed the comparison of agronomic traits and nutritional quality of the same varieties in saline-alkali land and non-saline-alkali land. Studies have shown that the agronomic traits and nutritional quality of the same cultivated oat varieties in non-saline-alkali land are much higher than those of oats cultivated in saline-alkali land.

L77: Yield and trait data are presented with means and significance indicators, but the specific statistical tests used are not clearly specified. Please do that and consider this during all results section where is the case

Re: Our statistical test method is LSD multiple comparison statistical method.

General comment: The results in my opinion are densely presented which may overwhelm readers, I recommend their reconsideration and those extensive or less relevant to be moved at supplementary materials. Please increase the font sizes from insides of the graphs

Re: We have modified the relevant content in the manuscript. (line 454- line 458)(Line 459-line 463)

L508: The discussion claims that oats are "recognized as a moderately salt-tolerant crop and can be utilized to enhance saline-alkali land" but for me there are not enough referencing specific prior studies or providing a comprehensive comparison of salt tolerance levels among different crops, that could strengthen the claim

Re: We have added previous studies to the discussion to prove that oat adaptability decreases with the increase of stress level (line 576-line 580).

L509: The statement that "the successful introduction of oats is primarily determined by the completeness of their growth and development" is in my opinion is very simplistic and lacks depth; it overlooks other critical factors such as genetic variability, soil conditions, management practices, and environmental stresses. This should be considered

Re: We mentioned in the discussion that different oat varieties have different genetic characteristics and different adaptability to the environment. The essence of the success of oat introduction is the dynamic balance regulation of the four-dimensional system of ' genetics-soil-management-environment '. However, ' growth and development integrity ' should be the ultimate goal.

L519: The comparison of yield reductions under saline-alkali conditions with previous studies (e.g., Bale Fresh Grass yield) is somewhat superficial as it does not account for differences in experimental conditions, soil salinity levels, or management practices, which limits the validity of the direct comparison

Re: thank you very much for your in-depth comments on the comparative analysis of yield reduction rate under saline-alkali conditions! The differences in experimental conditions, soil salinity gradients, and the impact of management measures that you pointed out have indeed hit the key limitations of current research. Due to the very few research reports similar to this study, it is difficult to quote references under the same test conditions and management measures. Therefore, we next prepare a comparative test related to this experiment. This experiment belongs to the cultivation of oat varieties in saline-alkali soil. The same oat varieties were compared with different experimental conditions by citing references ( different experimental conditions refer to: saline-alkali soil conditions in this experiment and non-saline-alkali soil conditions in the references).

L527: The assertion that "the salt accumulation in Mulesi reached 927.13 kg·hm-2 under identical conditions, surpassing Qinghai 444 by 515.10 kg·hm-2" implies a positive correlation between salt accumulation and yield without discussing potential thresholds or the possibility that excessive salt accumulation could be detrimental in my opinion

Re: In view of the problem of salt accumulation you mentioned, because there is no reference for salt accumulation in field experiments, oat salt accumulation = total salt content in aboveground plants × hay yield, so it has a greater advantage in hay yield, or higher total salt content in plants can achieve greater salt accumulation. There is no relevant research on the specific threshold. In view of the possibility that excessive salt accumulation may have adverse effects, the total salt content in plants in this study is expressed by crude ash. Crude ash is the residual inorganic mineral residue of forage grass after full combustion at high temperature of 550-600 °C, which is called 'mineral 'or 'inorganic '. It is not a single substance, but a mixture of oxides and salts of various inorganic elements, including calcium, phosphorus, potassium, magnesium, sodium, chlorine, sulfur and other major elements, iron, zinc, copper, manganese and other trace elements. The crude ash in forage is also an important way for herbivores to obtain minerals.

L533: please replace "li et al., " with "Li et al., "

Re: We have made the corresponding replacements in the article. (line 624)

L541-558: In my opinion the explanation for lower crude protein and RFV values in this study compared to previous literature is somewhat speculative. It attributes the decline mainly to "adverse effects of saline-alkali soil" without providing direct evidence or discussing possible methodological differences

Re: Thank you very much for your suggestion. We think your suggestions are very valuable for our article. The explanation for the low crude protein and RFV values in this study is speculative to some extent. This is indeed our guess, we will design new experiments to prove our speculation in the future.

L551: check "li et al.[22]."

Re: We have made the corresponding replacements in the article. (line 624)

L560: The discussion on soil physicochemical properties notes a decrease in pH, EC, and total salt content over the growth period, suggesting these are beneficial effects. However, it does not sufficiently address whether these reductions are sustainable or whether they could be transient or due to experimental artifacts

Re: The decrease of soil pH, EC and total salt content during the growth period may be the result of plant absorption, microbial activity and environmental factors. The current deficiency is the lack of long-term data to verify its sustainability. Due to the limited time and experimental funds, only the soil at the seedling stage and the harvest stage was used for comparison, which proved that the soil physical and chemical properties at the harvest stage were developing in a good direction. We will next combine long-term field experiments, multi-dimensional mechanism analysis and controlled studies to verify its sustainability.

L564: In my opinion the claim that "the reduction in soil salinity was greater in the 15-30 cm soil layer" is made without discussing the mechanisms behind this pattern, such as leaching, plant uptake, or soil processes, which would add depth to the interpretation

Re: We have added the relevant mechanism discussion in this section. (line 639-line 648)

L569: The statement that "the soil EC mostly exceeded 4 dS/m, categorizing them as saline-alkali land" is presented as a fact, but it lacks discussion on the implications for crop growth, thresholds for different crops, or how these levels compare to known salinity tolerance limits

L572: The discussion suggests that "the primary mechanism of salt tolerance might be osmotic stress tolerance" based on observed reductions in soil salt content, but it does not critically evaluate alternative mechanisms such as ion exclusion or tissue tolerance, nor does it cite relevant mechanistic studies

Re: We have re-discussed this part. (line 658-line 664)

L587: The discussion on K+ and Na+ levels attributes their patterns to "competitive inhibition" without referencing specific studies or mechanisms, and it oversimplifies the complex ion uptake and transport processes involved in salt stress responses

Re: We have re-discussed this part. (line 683-line691)

L594: The explanation of HCO3- reduction and its role in plant growth is brief and lacks detailed mechanistic insight or references to relevant physiological studies in my opinion

Re: We have re-discussed this part.(line 697-ling704)

General comments: In my opinion to build all the discussion on comparison/correlating the obtained research data with a single article "Li et al.," is very scarce diminishing seriously the scientific relevance. On other way, the discussion tends to be descriptive rather than critically analytical, often attributing observed phenomena to general concepts without sufficient mechanistic or literature support, reducing the scientific rigor of the interpretation in my view.

Re: We have revised the discussion section and cited relevant mechanisms and literature support. All the discussions on the comparison / association of the obtained research data with a single article ' Li et al. ' mentioned by the expert are very scarce. The fresh grass and hay yield in the discussion is cited by zhang et al. [1]. Flag leaf length, flag leaf width and flag leaf area are inseparable as a whole, because the leaves are rich in crude protein, so the study of LI et al. [2] was cited as a comparison ; rFV is an important index to select high-quality forage by using the comprehensive performance of ADF and NDF. It is widely used in the evaluation model of forage feeding value. This paper quotes Li et al [3]. Li et al. [4] was cited for the base ions in the soil. [ 4 ] For the experts, the discussion section you mentioned only cited a single LI et al.to discuss the problem, in fact, more than one reference, may be the same surname caused your misunderstanding.

Reference:

[1] R, Zhang.; Z, Xu.; W, Wang.; J, Ren.; JF, Tian.; L, Liu.; XL, Ma. Evaluation of Production Performance and Adaptability of 11 Oat Varieties in Diqing Region. Chinese Journal of Grassland 2023, 45, 32-40.

[2] HF, Li.; BW, Zhou.; M, Zhang.; SN, Shi.; ZJ, Li. Evaluation of Introduction Adaptability of Different Oat Varieties in Hulunbuir Region. Acta Prataculturae Sinica 2024, 33, 60-72.

[3] J, Li.; M, Nan.; YM, Liu.; CJ, Zhang.; SL, Ren; F, Bian. Comprehensive Evaluation of Yield, Quality and Feeding Performance of Different Oat Varieties. Acta Agrestis Sinica 2023, 31, 1089-1098.

[4] Li, R.; Shi, F.; Fukuda, K.; Yang, Y. Effects of salt and alkali stresses on germination, growth, photosynthesis and ion accumulation in alfalfa (Medicago sativaL.). Soil Science and Plant Nutrition 2010.

Round 2

Reviewer 1 Report

Comments and Suggestions for Authors

İt is ok

Author Response

Thank you very much for your review.

Reviewer 2 Report

Comments and Suggestions for Authors

Dear Authors,

Thank you for considering previous observations in "Comparative Study on Production Performance of Different Oat Varieties and Soil Physicochemical Properties in Saline-Alkali Land of Qaidam Basin" manuscript. Reading carefully this new version of the manuscript I have the following recommendation to it:

  1. At introduction, the manuscript fail to explicitly delineate the specific knowledge gap that the study addresses. While the manuscript mention "few reports on oats in the saline-alkali land of the Qaidam Basin," this don't adequately synthesize existing literature to demonstrate precisely what remains unknown. This in my opinion is a serious gap (L96)
  2. Similarly, the introduction presents salt stress effects (osmotic stress, ionic stress) without adequately explaining the physiological mechanisms specific to oats which are one of the main subject of this research. The molecular and biochemical pathways of salt tolerance in oats are not sufficiently elaborated (L44..)
  3. L755-757: explain the coefficients (88.9, ..., 1.29)
  4. Authors should check their text for editing, as there are numerous errors related to capitalization, numbers, etc.
  5. Please consider to better highlight the justification for the study's focus on this particular crop through highlighting the economic and agricultural significance of oats specifically for the Qaidam Basin region - present quantitatively/numerical estimation/data in discussion section
Comments on the Quality of English Language

Corrections and improvements are needed

Author Response

Point-by-point response to the reviewer comments

Dear Reviewers:

Thank you for your thoughtful suggestions and insights, which have benefited from the manuscript. I am looking forward to working with you to move this manuscript closer to publication in “Plants”.

The manuscript has been rechecked and the necessary changes have been made in accordance with your suggestions. The responses to all comments have been prepared and attached below. We have tried our best to solve the problems you proposed, and we hope that the revised manuscript is now suitable for publication in the journal “Plants”. If you have any questions remained about this paper, please feel free to contact us.

  1. At introduction, the manuscript fail to explicitly delineate the specific knowledge gap that the study addresses. While the manuscript mention "few reports on oats in the saline-alkali land of the Qaidam Basin," this don't adequately synthesize existing literature to demonstrate precisely what remains unknown. This in my opinion is a serious gap (L96)

Re: Thanks to the expert for your suggestions. This part has been revised again, and relevant references have been cited. (line 129-line 132) Regarding the issue that the paper suggested by you failed to clearly explain the specific knowledge gap to be addressed in this study, it is mentioned in the last paragraph of the introduction that the ultimate goal of this experiment is to screen out oat varieties suitable for planting in the saline - alkaline land of the Qaidam Basin. It belongs to the gap in special regions and research methods. Existing introduction experiments of oats in other research areas mainly rely on a single yield, or add other indicators of plants to screen oat varieties suitable for planting in other regions. This paper focuses on planting and cultivating oat varieties on saline - alkaline land, adding indicators such as soil physical and chemical properties, evaluating the adaptability of oats through the production performance of above - ground plants and the physical and chemical properties of underground soil, and screening oat varieties suitable for specific climates through the combination of production performance and soil physical and chemical properties. The research in this area at home and abroad is in a blank state. This experiment combines production performance with soil physical and chemical properties to screen out high - yield and high - quality forage varieties suitable for the climate and soil conditions in the Qaidam area. This also has certain reference significance for our later saline - alkaline land improvement experiments and selecting good varieties to participate in the experiments.

  1. Similarly, the introduction presents salt stress effects (osmotic stress, ionic stress) without adequately explaining the physiological mechanisms specific to oats which are one of the main subject of this research. The molecular and biochemical pathways of salt tolerance in oats are not sufficiently elaborated (L44..) (line 58-line 70) (line 110-line 125)

Re: Special thanks for your suggestions. Your suggestions are of great reference value for our article. We have revised this part of the content again, added relevant content, and cited related references.

  1. L755-757: explain the coefficients (88.9, ..., 1.29)

Re: Thank you very much for your review.88.9 is an intercept term (constant term) obtained through the statistics and fitting of a large amount of experimental data. 1.29 is a constant (correction coefficient) determined through the statistics and fitting of a large amount of data. It is a correction coefficient derived from the experimental data of a large number of roughages such as alfalfa (initially used for alfalfa and later extended to the evaluation of other roughages) through regression analysis and standardization processing. Referring to domestic and foreign literatures, this is a fixed formula. The relevant reference [69] (line 837) has been cited in the main text, and references [2] and [3] have been supplemented in the Cover letter.

  1. Authors should check their text for editing, as there are numerous errors related to capitalization, numbers, etc.

Re: We have checked and revised the full text, thank you very much.

  1. Please consider to better highlight the justification for the study's focus on this particular crop through highlighting the economic and agricultural significance of oats specifically for the Qaidam Basin region - present quantitatively/numerical estimation/data in discussion section

Re: We have added the relevant content to the manuscript. Please review it. (line 620-line 625)

  1. Comments on the Quality of English Language

Corrections and improvements are needed

Re: We have sought professional English speakers to correct and improve the English language quality of the full text.

References

[2] Cover crop systems impact on biomass production, carbon-to-nitrogen ratio, forage quality, and soil health in a semi-arid environment

[3] Li Jing, Nan Ming, Liu Yanming, et al. Comprehensive Evaluation of Yield, Quality and Feeding Performance of Different Oat Varieties[J]. Acta Agrestia Sinica. ,2023,31(4): 1089-1098
